# Evaluating the impact of multivariate imputation by MICE in feature selection

**Maritza Mera-Gaona** [1]*, **Ursula Neumann**[2], **Rubiel Vargas-Canas**[1], **Diego M. López**[1]

**1** University of Cauca, Colombia, Popayán, Cauca, Colombia, **2** Data Science and Optimization, Fraunhofer Center for Applied Research on Supply Chain Services SCS, Nuremberg, Bayern, Germany

* maritzag@unicauca.edu.co

**Data Availability Statement:** The breast-Cancer dataset is available at https://archive.ics.uci.edu/ml/datasets/Breast+Cancer+Wisconsin+(Diagnostic). The Letter-recognition dataset is available at https://archive.ics.uci.edu/ml/datasets/Letter

## Abstract

Handling missing values is a crucial step in preprocessing data in Machine Learning. Most available algorithms for analyzing datasets in the feature selection process and classification or estimation process analyze complete datasets. Consequently, in many cases, the strategy for dealing with missing values is to use only instances with full data or to replace missing values with a mean, mode, median, or a constant value. Usually, discarding missing samples or replacing missing values by means of fundamental techniques causes bias in subsequent analyzes on datasets. **Aim**: Demonstrate the positive impact of multivariate imputation in the feature selection process on datasets with missing values. **Results**: We compared the effects of the feature selection process using complete datasets, incomplete datasets with missingness rates between 5 and 50%, and imputed datasets by basic techniques and multivariate imputation. The feature selection algorithms used are well-known methods. The results showed that the datasets imputed by multivariate imputation obtained the best results in feature selection compared to datasets imputed by basic techniques or non-imputed incomplete datasets. **Conclusions**: Considering the results obtained in the evaluation, applying multivariate imputation by MICE reduces bias in the feature selection process.

## Introduction

Missing data is a common problem in real-world datasets. Even if the researchers work hard to avoid them, missing values frequently occur for different reasons. Consequently, missingness can lead to issues in analyzing the data because most statistical methods and packages exclude subjects with any missing value. The result is that analyzes are made only with complete cases, affecting precision and leading to biased results. Although removing incomplete data is a fast and straightforward technique, it is also a risky solution since in applying it we must assume that discarded data does not influence the dataset. As a result of discarding cases with missing values, datasets could lose many instances of interest [1].

Considering the above, before deciding how to handle missing values in a dataset, the researchers must determine what the missing values depend on. The choice of a correct

+Recognition. The Statlog – (Heart) dataset is available at https://archive.ics.uci.edu/ml/datasets/ statlog+(heart). The Spambase dataset is available at https://archive.ics.uci.edu/ml/datasets/ spambase. These are all third party data. The third-party data are available for everyone and does not require privileges to be accessed.

**Funding:** The work was funded by a grant from Colciencias, Colombian Agency of Science, Technology, and Innovation, under Funding call 647- 2015, project: "Mechanism of selection of relevant features for the automatic detection of epileptic seizures"; the funder provided support in the form of a scholarship for MMG but did not have any additional role in the study design, data collection and analysis, decision to publish, or preparation of the manuscript. Additionally, University of Cauca and Fraunhofer Center for Applied Research on Supply Chain Services SCS provided support in the form of salaries for DML and RVC, and UN, respectively. However, the employers did not have any additional role in the study design, data collection and analysis, decision to publish, or preparation of the manuscript. The specific roles of these authors are articulated in the 'author contributions' section.

**Competing interests:** The academic and commercial affiliations of the authors, Colciencias, Fraunhofer Center for Applied Research on Supply Chain Services SCS and University of Cauca, do not alter their adherence to PLOS ONE policies on sharing data and materials.

strategy will ensure an appropriate dataset to support subsequent analyzes such as Feature Selection and Classification.

According to Rubin [2,3] there are three types of mechanisms of missing values: (i) Missing Completely At Random (MCAR), (ii) Missing At Random, and (iii) Missing Not At Random (MNAR). Missingness is MCAR if the probability of having missing data does not depend on the observed data or missing variables. For example, when a sensor's battery runs out, the sensor stops sending data to servers. Missing data is called MAR when the missing values (values can be missing or not) are related to other available information but not on unobserved data, which means that some variables depend on others. An example is that women usually avoid revealing their age in surveys (gender is related to missingness in the age variable). MNAR occurs if the probability of missingness depends on the values of unobserved variables. For example, people with high salaries avoid revealing their incomes in surveys. For some researchers, the mechanisms of MAR and MNAR are similar and indistinguishable [4].

Many studies have been carried out in order to explore mechanisms for handling missing values in different fields [5–13]. Although choosing the method may be difficult, most studies conclude that imputation is better than removing data due to the fact that deleting data could bias datasets as well as subsequent analyzes on these [14]. Consequently, data imputation is an important preprocessing task in Machine Learning.

An additional problem in the last few years is the proliferation of datasets with hundreds or even tens of thousands of variables. Thus, feature selection (FS) has become an option for reducing high dimensionality, redundant features, or noise from datasets [15]. Nevertheless, in real scenarios it is necessary to deal with missing values in the datasets and the most common FS techniques consider only datasets with complete data in the independent variables.

According to [16], missing values could be present in the target variable in the classification context. For example, when a classification or estimation model is evaluated, missing values are imputed in the test data's target variable and the model predicts values for the target variable. However, when a dataset has missing values in the features, we must find a way to handle the missing values and perform preprocessing tasks to get a dataset with complete data. Commonly, the missing data problem is solved by removing the instances or features with missing values or replacing the missing values using basic mechanisms such as mean, mode, etc. Although these strategies are easy to implement, they change the distribution of the datasets and may bias subsequent Machine Learning analyzes, for instance the feature selection or classification processes. On one hand, the methods to handle missing values could eliminate from the dataset: (i) relevant features or (ii) instances that reveal the importance of the relevant features. On the other hand, the machine learning models could be trained using only a part of the original datapoints.

Some studies have proposed new techniques to carry out FS on datasets with missing values [17–19]. Although these studies showed promising results, the authors' experiments did not evaluate the effect of data imputation on datasets to analyze whether or not the imputation methods bias the FS process. Moreover, the experiments in [17] and [19] were carried out using only rates of missing values less than or equal to 10%.

In previous studies, we evaluated how feature selection improved the performance of the classification of epileptic events and normal brain activity in Electroencephalograms [20,21]. The experiments were carried out using datasets with high dimensionality in a scenario with the need of reducing the computational complexity. The results indicated that the best subset of relevant features was selected by an approach based on Ensemble Feature Selection (EFS).

We thus proposed a Framework of Ensemble Feature Selection to improve the selection of relevant features in datasets with high dimensionality [22]. Nonetheless, one of the weakness of the original proposal was the handling of datasets with missing values. In the real world,

datasets have a high probability of having incomplete data, which means that handling missing values is necessary before selecting relevant features. This renders the results of FS uncertain when the dataset has incomplete data.

This research aims to describe how data imputation can improve feature selection on datasets with missing data and avoid biasing the dataset. For this, we showed the impact of missing values in the FS process by implementing a data imputation algorithm and evaluating it with different datasets to compare the FS process using datasets without handling missing values versus imputed datasets. In light of this, this paper is organized as follows: Section 2 presents the datasets used to evaluate our proposal and theoretical descriptions about basic mechanisms for handling Missing Values, Multivariate Imputation, Multiple Imputation, and Feature Selection. In Section 3, the evaluation and results are presented. Section 4 describes the discussion of results. Finally, the main conclusions are laid out in Section 5.

## Materials and methods

### Systematic mapping studies in software engineering

To review works related to FS and data imputation, we carried out two systematic mappings focused on identifying studies related to imputation and the assembly of feature selection algorithms following the guidelines described by Petersen [23]. We used two search strings, one for each topic:

- Imputation data: (imputation data) and (missing values or missingness rates or incomplete data or incomplete dataset)

- Feature selection: ("framework" and "ensemble") and ("dimensionality reduction" or "feature selection") and ("EEG" and "automatic") and ("detector" or "reading" or "recognition" or "analysis").

The searches guided by the previous keywords, were used to find relevant papers from IEEE, PubMed, and Science Direct databases. The analysis of the papers was led following review criteria based on the quality of their contributions, particularly the proposal of imputation and assembly of feature selection algorithms.

### Datasets

This research uses 4 datasets [24–27], *Breast-cancer*, *letter-recognition*, *Statlog—Heart and Spambase*, from UCI Machine Learning Repository [28] to evaluate our proposal. These collections include categorical and numerical features and contain data from different fields. Besides, the datasets are available for everyone and do not require privileges to be accessed.

*The Breast-Cancer* dataset contains data provided by the Oncology Institute [24]. Each instance is described by 9 attributes and represents information from a patient.

*Letter-recognition* is a dataset that represents 26 capital letters in the English alphabet [25]. The dataset was built considering the black-and-white pixel representation on 20 different fonts. Each representation was randomly distorted to get 20.000 instances. Each instance was converted into 16 numerical features.

The *Statlog–(Heart)* dataset contains information about heart diseases. This dataset is a modified version of the *Heart Disease* dataset [26].

The *Spambase* dataset is a collection of spam and non-spam emails [27]. It is described by 57 attributes representing emails from emails classified as spam, work or personal emails.

Table 1 describes the number of categorical and numerical features and the number of instances in each dataset.

**Table 1. Datasets.**

| Dataset | Categorical | Numerical | Instances |
|---|---|---|---|
| Breast-Cancer | 9 | 0 | 286 |
| Letter-recognition | 0 | 16 | 20000 |
| Statlog—(Heart) | 7 | 6 | 269 |
| Spambase | 0 | 57 | 4601 |

**Removing data.** The most basic method for handling missing values in datasets is removing data. However, this option could delete all class instances, remove relevant variables, unbalance the dataset, and generate biases in classification or prediction.

**Listwise.** Listwise deletion removes all data for a case with at least one missing value. If the dataset contains a small number of instances, this strategy can remove all samples from one or more classes. Besides, when we remove the dataset cases, the result unbalances the dataset in most cases.

**Dropping variables.** Dropping variables is a good option when the variables with missing values are insignificant. Nonetheless, it is difficult to know the relevant features without making a feature selection analysis. Considering the above, imputation is usually better than dropping variables.

**Imputation.** Imputation allows replacing missing values with substitute or replacement values. There is a wide variety of imputation methods, and their main differences are associated with the process used to calculate the new values. It is relevant to mention that imputation does not necessarily give better results because a suitable imputation method cannot always be found.

**Mean, median and mode replacement.** A primary imputation method is to replace missing values with the overall mean, median, or mode. Although it is a fast strategy, this method presents clear disadvantages such as the mean, median, or mode imputation to reduce variance in the dataset.

**Multivariate imputation by chained equations.** Multivariate imputation by chained equations (MICE) is an imputation method based on Fully Conditional Specification, where different models impute incomplete attributes. Hence, MICE can impute missing values in datasets with continuous, binary, and categorical attributes by using a different model for each attribute. Thus, each attribute is modeled according to its distribution; for example, binary or categorical variables are modeled using logistic regression and continuous variables using linear regression. In the regression models, the modeled attribute represents the dependent variable, and the remaining attributes represent the independent variables. MICE algorithm considers the assumption that missing values are MAR, which means that its use in a dataset where the missing values are not MAR could generate biased imputations.

The MICE algorithm is described below.

1. Build a basic imputation for every missing value in the dataset.

2. Set back missing values for one feature ($F_x$).

3. The observed values of $F_x$ are used to train a prediction model in which $F_x$ is a dependent variable, and the other features are independent.

4. The missing values for $F_x$ are replaced with the predictions calculated by the model built in step 3.

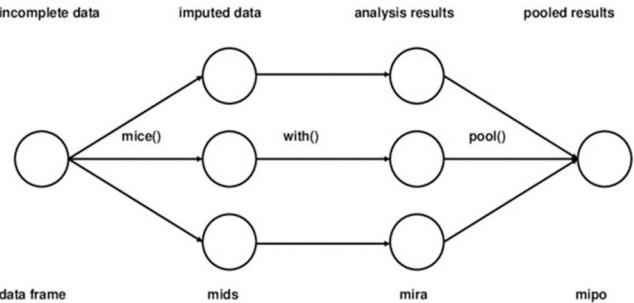

**Fig 1. Main steps used in multiple imputation [31].**

5. For each feature with missing values, steps 2–4 are repeated. When a prediction model has imputed all features with missing values, one cycle or iteration is finished.

6. Steps 2–5 are repeated for n iterations, and the imputations are updated at each cycle. The objective is to use the number of iterations to achieve a stable imputation. The imputed dataset is obtained in the last iteration.

The researcher determines the number of iterations $n$. Many iterations can improve imputation or promote overfitting. The stable number of iterations must be found by testing different values and depends on the data and missing values.

According to the MICE algorithm, we obtain one imputed dataset when the algorithm performs $n$ iterations. Additionally, if the previous process is repeated $m$ times, we get multiple imputed datasets.

**Multiple imputation.** Multiple imputation is a mechanism for creating multiple complete datasets in which for each missing value we calculate $m$ predictions [29]. The goal of multiple imputation is predicting or estimating the missing values and considering the uncertainty about missing values and the imputation model. This approach is not meant for generating new values only because a single unique value could be calculated using more straightforward means [30].

Fig 1 shows the main steps of Multiple Imputation.

MICE is a technique used to produce multiple imputations and pool them into one imputed dataset [32]. The standard strategy in Multiple Imputation is building a large joint model to predict all attributes with missing values. However, this approach is challenging to implement when there are hundreds of variables of different types. In these cases, MICE is an excellent option for handling the types [33], since the algorithm establishes a series of regression models according to the distribution and type of each attribute.

## The setting of multiple imputation by MICE

• Number of Imputations

A critical task in Multiple Imputation is defining the number of datasets that we must impute. All imputed datasets contain the same data according to the original observed data; the differences appear initially with only the missing values. The literature recommends the number of imputed datasets ought to be between 5 and 10 [29].

• Data to train the prediction models.

A relevant aspect to consider in setting up MICE is selecting the variables or attributes included in the imputation process. Usually, we use all available variables, especially those used in subsequent analyses such as feature selection and classification/estimation. In [29], the authors consider three important points in selecting variables and their values: (i) the imputation model must be more general than the analysis model; then, if it is possible, including "auxiliary" variables (in the imputation regression model of a variable) that will not be used in the analysis process but offer information to improve the imputations; (ii) Defining whether the imputations are calculated at the item level or the summary level; for example, when there are variables constructed from other variables, it is necessary to decide if it is better to impute the original variables or the resulting variables; and, (iii) determining if the imputations will be calculated to reflect raw scores or standardized scores.

In some cases, researchers have proposed using outcome-dependent variables in the imputation model to include all possible relationships in the imputation regression model [34]. This assumption is based on the fact that the outcome depends on variables to impute. If outcomes are excluded from the imputation process, imputations will be calculated assuming that these are independent of the outcome.

- Pooling

The *m* imputed datasets generated by multiple imputation are pooled considering the types of attributes with missing values in the dataset. For instance, binary or categorical attributes are usually pooled, finding the mode of predictions and numerical attributes, calculating the mean of predictions [31].

## Feature selection

**Select K Best.**   Select K Best (SKB) is an FS algorithm for selecting a set of features according to the *k* highest scores. Scores are calculated using a test between each feature and the target. Some of the most widely used tests are described below.

**Chi-squared.**   Chi-squared is a statistical test to evaluate features and determine whether these are dependent or independent of the target. If a feature is independent, it is considered irrelevant to the classification. Eq 1 describes the Chi-squared test.

$$X^2 = \frac{(Observed\ frequency - Expected\ frequency)^2}{Expected\ frequency} \tag{1}$$

Where *observed frequency* is the number of class observations and *expected frequency* the number of expected class observations if there was no relationship between feature and target.

**F-test and ANOVA F-test.**   These are statistical tests to evaluate features and obtain the significance of each feature to improve a classification or regression model. The result of these measures is a subset of features with the *k* most informative features.

**Recursive feature elimination.**   The RFE algorithm uses an external estimator to evaluate the importance of the features. Recursively, it removes features and evaluates the remaining subset by building a model with the current subset of features. The accuracy of the model is used to identify which features contribute to improving the prediction. The algorithm thus eliminates the worst-performing features on a model until the best subset is found.

**Feature importance measures for tree models.**   The importance of a feature is calculated using Decision Trees, or the ensemble methods built upon them. One of the most common measures is Gini importance [35], based on the impurity reduction of splits. This counts when a feature is used to split a node, weighted by the number of samples it divides. When a tree model is trained using scikit-learn [36], a vector with the importance of each feature is

calculated. The sum of the vector values is 1. Vector values can be used as scores to select the k most essential features, where the feature with the highest score is the most important.

**Metrics to evaluate imputation methods.** We calculated the mean absolute error (MAE) and root mean square error (RMSE) between imputed values and original values in numerical variables and accuracy in categorical variables to evaluate the imputation quality.

- MAE and RMSE

The mean absolute error and the root mean square error are the standard statistical metrics used to evaluate models [37].

MAE and RMSE are described by Eqs 2 and 3,

$$MAE = \frac{1}{n}\sum_{i=1}^{n}|e_i| \tag{2}$$

$$RMSE = \sqrt{\frac{1}{n}\sum_{i=1}^{n}e_i^2} \tag{3}$$

where $e_i$ represents $n$ samples of model errors ($e_i$, $i = 1, 2, \ldots, n$). To evaluate the quality of imputations, we considered Eqs 4 and 5. Where $\hat{Y}_i$ represents the values predicted by imputation and $Y_i$ real values.

$$MAE = \frac{1}{n}\sum_{i=1}^{n}|\hat{Y}_i - Y_i| \tag{4}$$

$$RMSE = \sqrt{\frac{1}{n}\sum_{i=1}^{n}(\hat{Y}_i - Y_i)^2} \tag{5}$$

- Accuracy

Accuracy is an error-rate used to evaluate the performance of classification models. It estimates the overall probability of correct classification of a test sample [38]. Accuracy is described by Eq 6,

$$error = \frac{FN + FP}{N} \tag{6}$$

where $N$ is the total of instances, $FN$ the number of false negatives, and $FP$ the number of false negatives.

## Results

In this section, we present the evaluation results for analyzing the quality of imputation and the behavior of the process of feature selection on datasets imputed by MICE and mean/mode replacement.

### Evaluating the quality of imputation

The described datasets were used to create simulated realistic datasets with missing values. Each original dataset was transformed considering 10 levels of missing data (% missingness = 5, 10, 15. . .,45, and 50), and for each level, the transformation was repeated 100 times. Hereafter

we refer to datasets with randomly removed missing values as *simulated datasets*. Besides, each simulated dataset was imputed using MICE and mean/mode replacement.

Once the imputed datasets were generated and processed, we compared them with the original datasets to evaluate the quality of the imputations.

**Outcomes.** The MICE algorithm was evaluated comparing the imputed values with real values in the original dataset. We further compared the imputation calculated by MICE with the imputation calculated by mean/mode replacement. The latter is the most common and basic solution implemented to impute missing values. For this, the simulated datasets were imputed 100 times with the two methods mentioned for each missingness rate. To evaluate if the imputed values were correct, we calculated MAE and RMSE for imputations in numerical variables and accuracy for categorical variables.

- Evaluation: Breast-cancer

Fig 2 describes the overall accuracy of imputations calculated by the MICE algorithm and mode imputation.

Fig 3 describes the accuracy by the feature of imputations calculated using the MICE algorithm.

Fig 4 describes the accuracy by the feature of imputations calculated using mode replacement.

Table 2 describes the overall accuracy of imputations calculated using MICE and mode replacement. According to the results, the overall accuracy achieved by MICE was better than the overall accuracy achieved by mode replacement in 100% of the missingness rates.

According to the results given in Tables 15 and 16 in Appendix A, the accuracy of the MICE imputation outperformed the accuracy of mode replacement in 97.59% of missingness rates by feature. Mode replacement obtained the best performance only for missingness rates of 35% and 40% in feature F3.

- Evaluation: Letter-recognition

Table 3 describes the overall MAE and RMSE of imputations calculated using MICE and mean replacement.

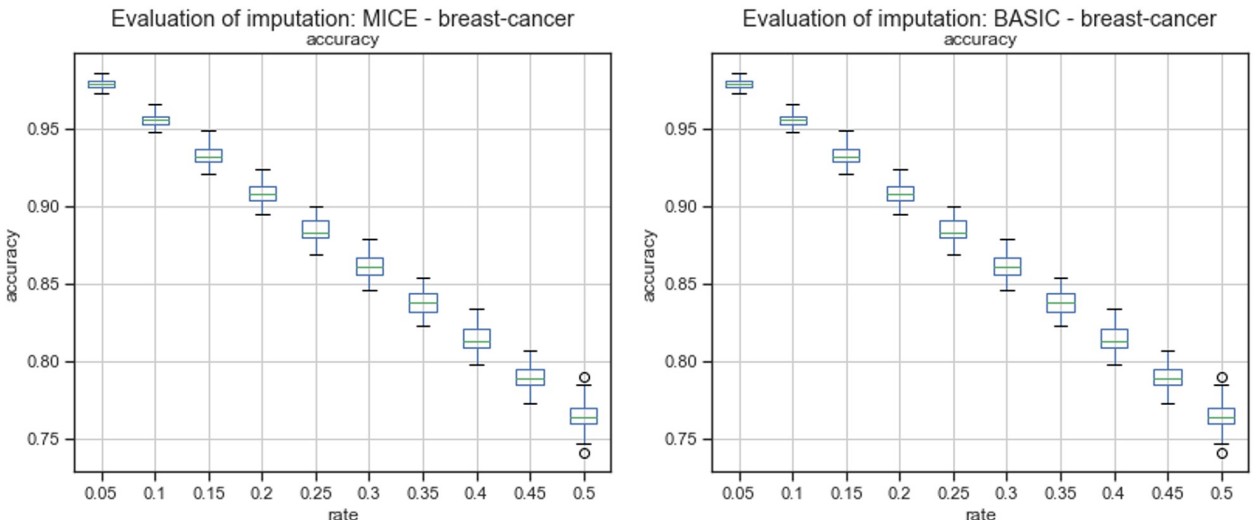

**Fig 2. Accuracy of imputations by MICE and mode.**

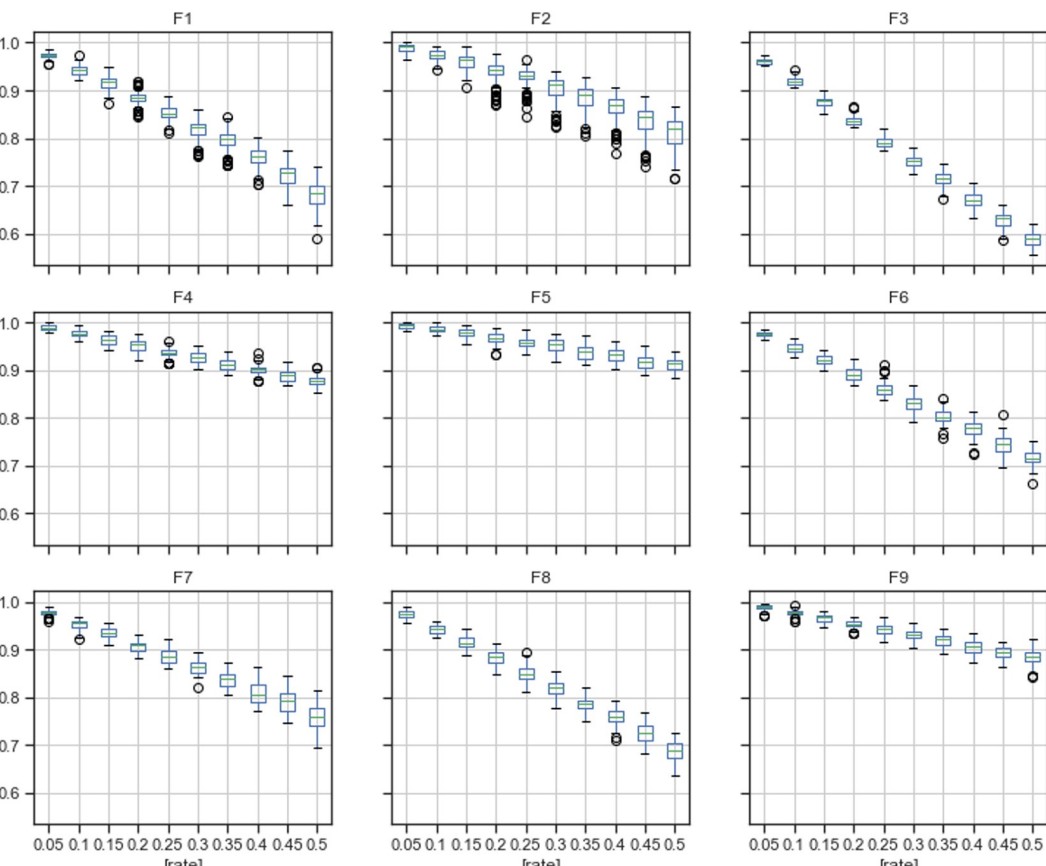

**Fig 3. Accuracy of imputations of MICE by feature.**

According to the results, the overall MAE and RMSE achieved by MICE were better than the overall MAE and RMSE achieved by mean replacement in 100% of missingness rates.

The MAE and RMSE of imputation calculated using MICE outperformed the MAE and RMSE of imputation calculated by mean replacement in 99.62% and 96.87% of the missingness rates by feature. Considering the number of variables of the letter-recognition dataset, we calculated these percentages but did not show each feature's results and each missingness rate.

- Evaluation: Statlog (heart)

Considering that the *statlog* dataset has categorical and numerical variables, we showed MAE and RMSE for numerical variables and accuracy for categorical variables.

Table 4 describes the overall accuracy of imputations calculated using MICE and mode replacement. According to the results, the overall accuracy achieved by MICE was better than the overall accuracy achieved by mode replacement in 100% of missingness rates.

According to the results given in **Tables 17** and **18** in Appendix B, the accuracy of MICE's imputation outperformed the accuracy of mode replacement in 75% of the missingness rates by feature.

Table 5 describes the overall MAE and RMSE of imputations calculated using MICE and mean replacement.

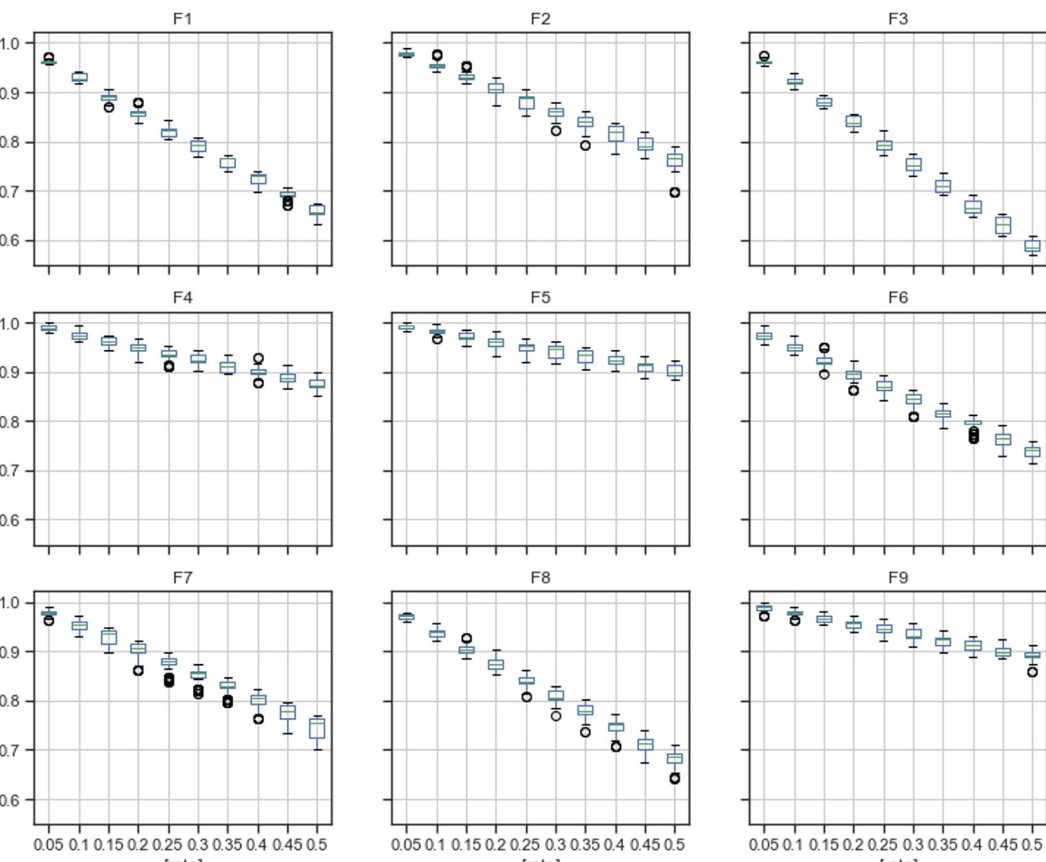

**Fig 4. Accuracy of mode imputations by feature.**

According to the results, the overall MAE and RMSE achieved by MICE were better than MAE and RMSE achieved by mean replacement in 100% of the missingness rates.

In accordance with the results given in **Tables 19** and **20** in Appendix B, the MAE of the imputation of MICE outperformed the MAE of mean replacement in 81.42% of the missingness rates by feature. Also, **Table 21** in Appendix E and **Table 22** in Appendix F show that the

**Table 2. The overall accuracy of MICE and mode.**

| RATE | MICE | MODE |
|------|------|------|
| 0.05 | 0.979 | 0.970 |
| 0.1 | 0.957 | 0.950 |
| 0.15 | 0.936 | 0.921 |
| 0.2 | 0.912 | 0.900 |
| 0.25 | 0.889 | 0.869 |
| 0.3 | 0.865 | 0.846 |
| 0.35 | 0.843 | 0.812 |
| 0.4 | 0.819 | 0.801 |
| 0.45 | 0.793 | 0.781 |
| 0.5 | 0.768 | 0.750 |

**Table 3. The overall MAE and RMSE.**

| RATE | MICE | | MEAN | |
|---|---|---|---|---|
| | MAE | RMSE | MAE | RMSE |
| 0.05 | 0.0773 | 0.1063 | 0.1171 | 0.1542 |
| 0.1 | 0.0798 | 0.1094 | 0.1171 | 0.1543 |
| 0.15 | 0.081 | 0.1109 | 0.1173 | 0.1544 |
| 0.2 | 0.0834 | 0.1138 | 0.1171 | 0.1542 |
| 0.25 | 0.0872 | 0.1187 | 0.1172 | 0.1544 |
| 0.3 | 0.0924 | 0.1256 | 0.1171 | 0.1541 |
| 0.35 | 0.0929 | 0.1263 | 0.1172 | 0.1542 |
| 0.4 | 0.0938 | 0.1271 | 0.1172 | 0.1542 |
| 0.45 | 0.0948 | 0.1279 | 0.1176 | 0.1544 |
| 0.5 | 0.0952 | 0.1283 | 0.1176 | 0.1544 |

**Table 4. The overall accuracy of MICE and mode.**

| RATE | MICE | MODE |
|---|---|---|
| 0.05 | 0.984 | 0.982 |
| 0.1 | 0.966 | 0.962 |
| 0.15 | 0.949 | 0.943 |
| 0.2 | 0.931 | 0.923 |
| 0.25 | 0.914 | 0.904 |
| 0.3 | 0.895 | 0.885 |
| 0.35 | 0.877 | 0.866 |
| 0.4 | 0.858 | 0.848 |
| 0.45 | 0.838 | 0.83 |
| 0.5 | 0.819 | 0.812 |

RMSE of the imputation of MICE outperformed the RMSE of the mode replacement in a 68.85% of the missingness rates by feature.

- Evaluation: Spambase

Table 6 describes the overall MAE and RMSE of imputations calculated using MICE and mean replacement.

**Table 5. The overall MAE and RMSE.**

| RATE | MICE | | MEAN | |
|---|---|---|---|---|
| | MAE | RMSE | MAE | RMSE |
| 0.05 | 0.141 | 0.189 | 0.173 | 0.217 |
| 0.1 | 0.142 | 0.191 | 0.175 | 0.22 |
| 0.15 | 0.145 | 0.195 | 0.174 | 0.22 |
| 0.2 | 0.146 | 0.198 | 0.174 | 0.22 |
| 0.25 | 0.152 | 0.205 | 0.174 | 0.22 |
| 0.3 | 0.156 | 0.212 | 0.174 | 0.22 |
| 0.35 | 0.162 | 0.22 | 0.174 | 0.22 |
| 0.4 | 0.167 | 0.226 | 0.175 | 0.221 |
| 0.45 | 0.168 | 0.226 | 0.174 | 0.221 |
| 0.5 | 0.168 | 0.225 | 0.174 | 0.221 |

**Table 6. The overall MAE and RMSE.**

| RATE | MICE | | MEAN | |
|------|------|------|------|------|
| | MAE | RMSE | MAE | RMSE |
| 0.05 | 0.0185 | 0.0508 | 0.0229 | 0.0568 |
| 0.1 | 0.0187 | 0.0509 | 0.023 | 0.0569 |
| 0.15 | 0.0189 | 0.0511 | 0.0231 | 0.0565 |
| 0.2 | 0.0195 | 0.0522 | 0.0234 | 0.057 |
| 0.25 | 0.02 | 0.0531 | 0.0234 | 0.0566 |
| 0.3 | 0.0215 | 0.0553 | 0.0237 | 0.0565 |
| 0.35 | 0.0234 | **0.0579** | 0.0241 | **0.0568** |
| 0.4 | 0.0233 | **0.0572** | 0.0241 | **0.0565** |
| 0.45 | 0.0239 | **0.0579** | 0.0247 | **0.0571** |
| 0.5 | 0.0241 | **0.0575** | 0.0249 | **0.0569** |

According to the results, the overall MAE and RMSE achieved by MICE outperformed the overall MAE and RMSE achieved by mean replacement in 100% and 60% of missingness rates, respectively.

The MAE and RMSE of imputation calculated using MICE outperformed the MAE and RMSE of imputation calculated by mode replacement in 77.36% and 70% of the missingness rates by feature, respectively. Considering the number of variables of the *spambase* dataset, we calculated these percentages but did not show each feature's results and each missingness rate.

**Densities.**   Figs 5–7 describe each variable's probability density function of the complete *breast-cancer* dataset and datasets imputed using MICE and mode replacement. According to

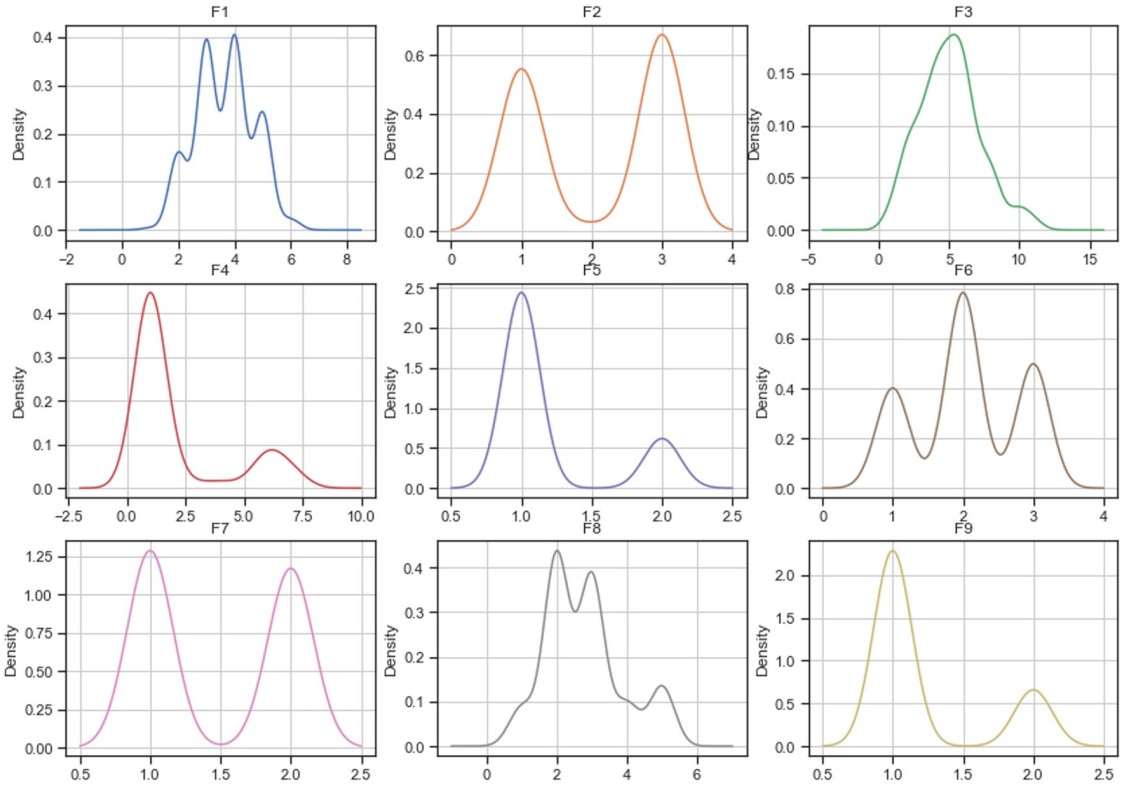

**Fig 5. Distribution of complete breast-cancer dataset.**

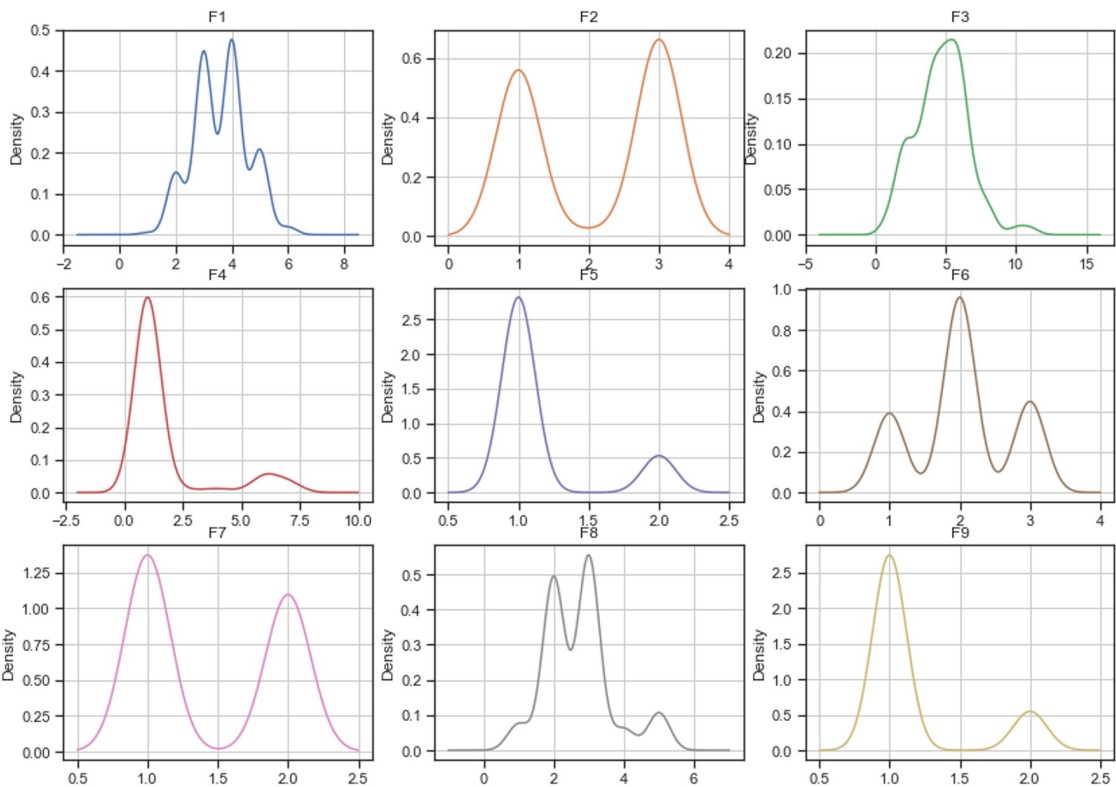

**Fig 6. Distribution of breast-cancer dataset imputed by MICE.**

the figures, the imputation calculated using MICE has densities similar to the complete dataset ones. However, most densities of datasets imputed using mode replacement did not only change in their shape but also increased the probabilities for some values compared to the complete dataset.

Considering the number of variables of the *startlog* (heart), *spambase*, and *letter-recognition* datasets, the densities of their variables are not shown in this paper. However, they were plotted and analyzed. As a result of this analysis, the imputations calculated by MICE maintain their densities close to the densities of the complete dataset, while densities of the imputed dataset using mode/mean replacement changed their shapes and probabilities.

## Evaluating feature selection

To evaluate the impact of missing values on feature selection, we simulated realistic datasets using the datasets described in Table 1. For each dataset, we generated three datasets with three different missingness rates: 25%, 30%, and 35%. Considering the simulated datasets, five FS algorithms were used to select relevant features on the complete dataset, on the dataset imputed using MICE, the dataset imputed using basic methods (Mean/Mode replacement), the dataset without missing values in instances (listwise elimination), and the dataset without missing values on variables (dropping variables).

**Letter-recognition.** Table 7 describes the *letter-recognition* dataset's relevant features that were selected using five algorithms of feature selection.

The results of applying five feature selection algorithms on datasets generated from simulations of missing values in the *letter-recognitio*n dataset are described in **Table 19** in Appendix

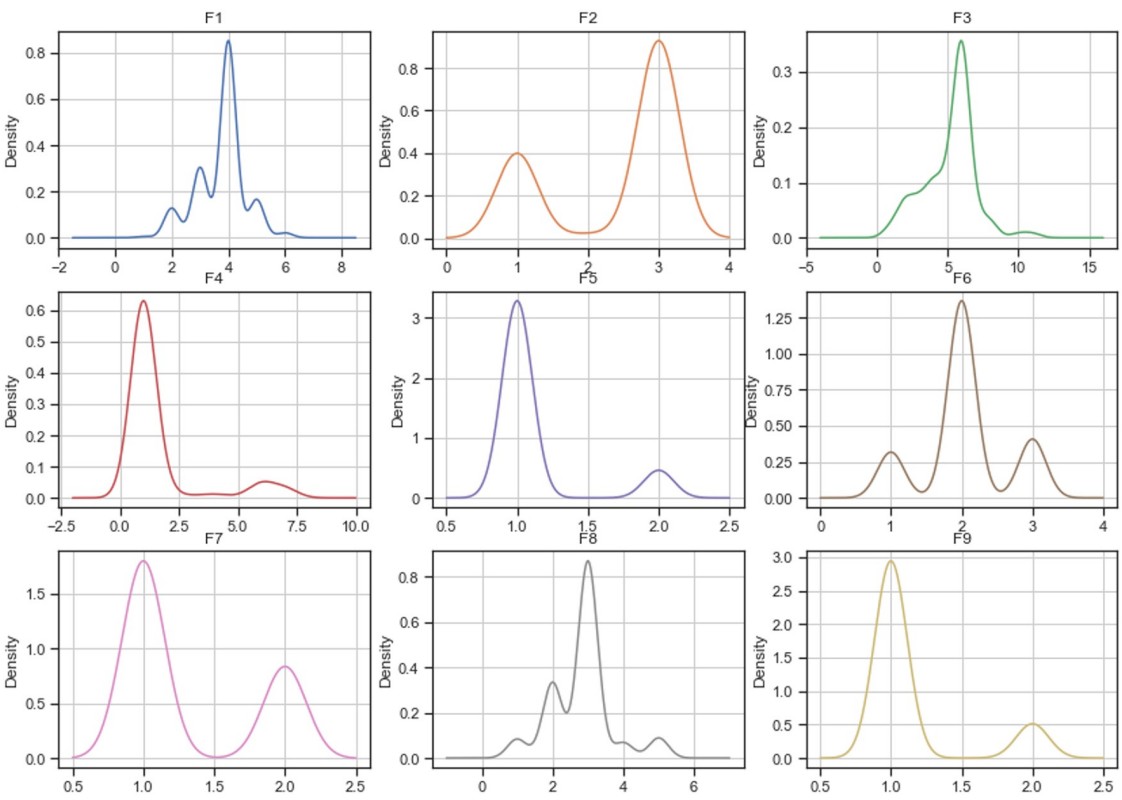

**Fig 7. Distribution of breast-cancer dataset imputed by mode.**

C. Each simulated dataset handled missing values with imputation by MICE and mean/mode replacement, listwise deletion, and dropping variables.

Fig 8 describes the intersection between the set of relevant features of *the letter-recognition* dataset and each simulated dataset's relevant features.

According to the results in Table 7 and Fig 8, the datasets imputed using MICE obtained the same set of relevant features as the complete dataset. The results also showed that datasets that were imputed using basic methods or removing instances of variables with missing values were influenced by dataset changes and produced different sets of relevant features.

**Statlog (heart).** Table 8 describes the relevant features of the *statlog* dataset that were selected using five algorithms of feature selection.

The results of applying five feature selection algorithms on datasets generated from simulations of missing values in the *statlog* dataset are presented in **Table 20** in Appendix D. Each simulated dataset handled missing values with imputation by MICE and mean/mode replacement, listwise deletion, and dropping variables.

**Table 7. Results of feature selection of the letter-recognition dataset.**

| dataset | Algorithm | |
|---|---|---|
| Full | Select K Best (Chi-squared) | F11, F13, F15 |
| | Select K Best (F-value) | F7, F11, F14 |
| | Select K Best (ANOVA F-value) | F7, F11, F13 |
| | Feature Recursive Elimination | F12, F13, F14 |
| | Feature Importance | F9, F13, F15 |

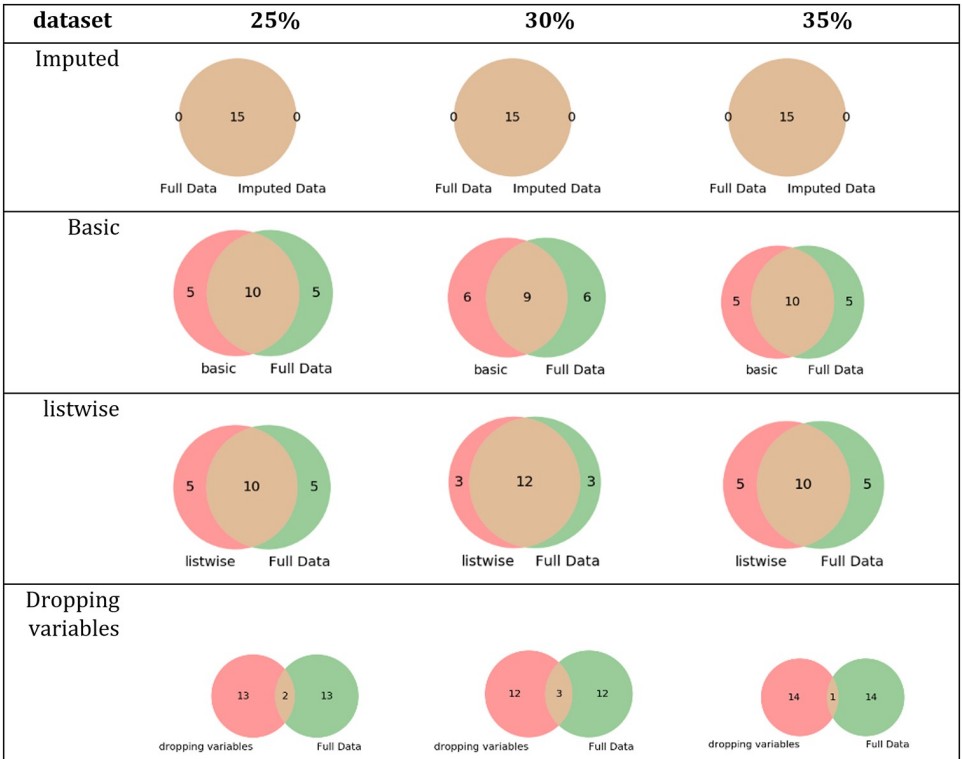

**Fig 8. Intersections of sets of relevant features of the letter-recognition dataset and its simulated datasets.**

Fig 9 describes the intersection between the set of relevant features of *the statlog* dataset and each simulated dataset's relevant features.

**Spambase.** Table 9 describes the relevant features of the *spambase* dataset that were selected using five algorithms of feature selection.

The application of five feature selection algorithms on datasets generated from simulations of missing values in the *spambase* dataset is shown in **Table 21** in Appendix E. Each simulated dataset handled missing values with imputation by MICE and mean/mode replacement, listwise deletion, and dropping variables.

Fig 10 describes the intersection between the set of relevant features of the *spambase* dataset and the relevant features of each simulated dataset.

**Breast-cancer.** Table 10 describes the relevant features of the *breast-cancer* dataset selected using five feature selection algorithms.

The results of applying five feature selection algorithms on datasets generated from simulations of missing values in the *breast-cancer* dataset are given in **Table 22** in Appendix F. Each

**Table 8. Results of feature selection of the statlog dataset.**

| dataset | Algorithm | |
|---------|-----------|---|
| Full | Select K Best (Chi-squared) | F3, F9, F12, F13 |
| | Select K Best (F-value) | F3, F9, F12, F13 |
| | Select K Best (ANOVA F-value) | F3, F9, F12, F13 |
| | Feature Recursive Elimination | F8, F10, F12 |
| | Feature Importance | F3, F9, F12, F13 |

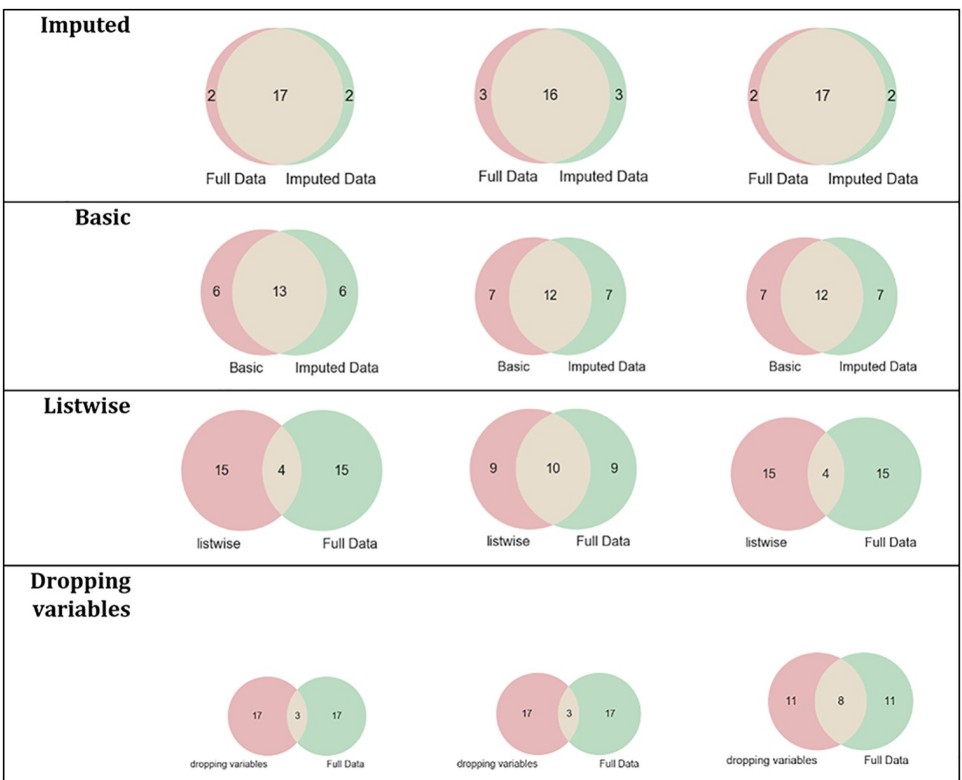

**Fig 9. Intersections of sets of relevant features of the statlog dataset and its simulated datasets.**

simulated dataset handled missing values with imputation by MICE and mean/mode replacement, listwise deletion, and dropping variables.

Fig 11 describes the intersection between the set of relevant features of the *breast-cancer* dataset and the set of relevant features of each simulated dataset.

## Discussion

In this work, we built an implementation of the MICE algorithm to evaluate the impact of multivariate and multiple imputation in datasets with categorical, numerical, and mixed categorical and numerical variables. The algorithm was assessed using datasets with different rates of missing values, which were generated randomly. The results were compared with the results of simple methods to handle missing values. The evaluation measured the quality of imputation, the distribution of imputed variables, and the impact in feature selection on imputed datasets.

To set up our MICE algorithm for each dataset, we took into account some aspects discussed in previous studies. For instance, Graham [39] suggests increasing the number of

**Table 9. Results of feature selection of the *spambase* dataset.**

| dataset | Algorithm | |
|---------|-----------|---|
| Full | Select K Best (Chi-squared) | F25, F27, F55, F56, F57 |
| | Select K Best (F-value) | F7, F19, F21, F23, F53 |
| | Select K Best (ANOVA F-value) | F7, F19, F21, F23, F53 |
| | Feature Recursive Elimination | F7, F27, F53 |
| | Feature Importance | F7, F16, F21, F52, F53 |

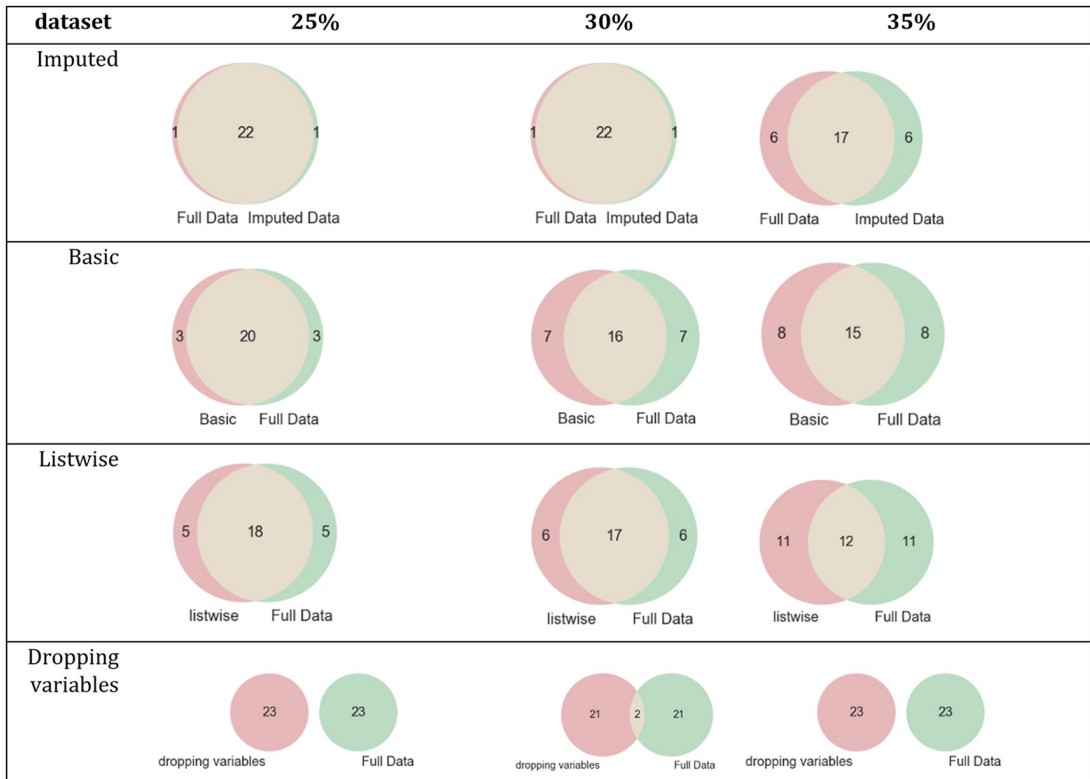

**Fig 10. Intersections of sets of relevant features of the spambase dataset and its simulated datasets.**

imputations to as many as 40 to improve imputation power when datasets have a high percentage of missing values. In practice, Graham also describes that many imputations could be inappropriate due to the dataset size, the models used to impute it, the amount of missingness in the data, and the available computer resources. In this sense, the imputation of a single dataset can take minutes, hours, or days. Thus, for datasets with hundreds or thousands of attributes and instances and a high rate of missingness, it would be impractical to calculate 40 imputed datasets as this could take hours or days. Consequently, we used many imputations for datasets with small sizes and smaller imputations for datasets of larger dimensions.

In accordance with the evaluation, the RMSE described in the previous section showed a good performance of all imputations calculated using MICE for all missingness rates. According to [40], a good result must be low (<0.3), and all results of RMSE of the MICE algorithm are less than 0.3 in overall results and results by feature.

The evaluation conducted in this paper was divided into two stages: reviewing of quality of imputation and analyzing results of FS on imputed datasets. For *the breast-cancer* dataset, the

**Table 10. Results of feature selection of the breast-cancer dataset.**

| dataset | Algorithm | |
|---|---|---|
| Full | Select K Best (Chi-squared) | F3, F4, F5, F6 |
| | Select K Best (F-value) | F4, F5, F6, F9 |
| | Select K Best (ANOVA F-value) | F4, F5, F6, F9 |
| | Feature Recursive Elimination | F5, F6, F7 |
| | Feature Importance | F1, F3, F6, F8 |

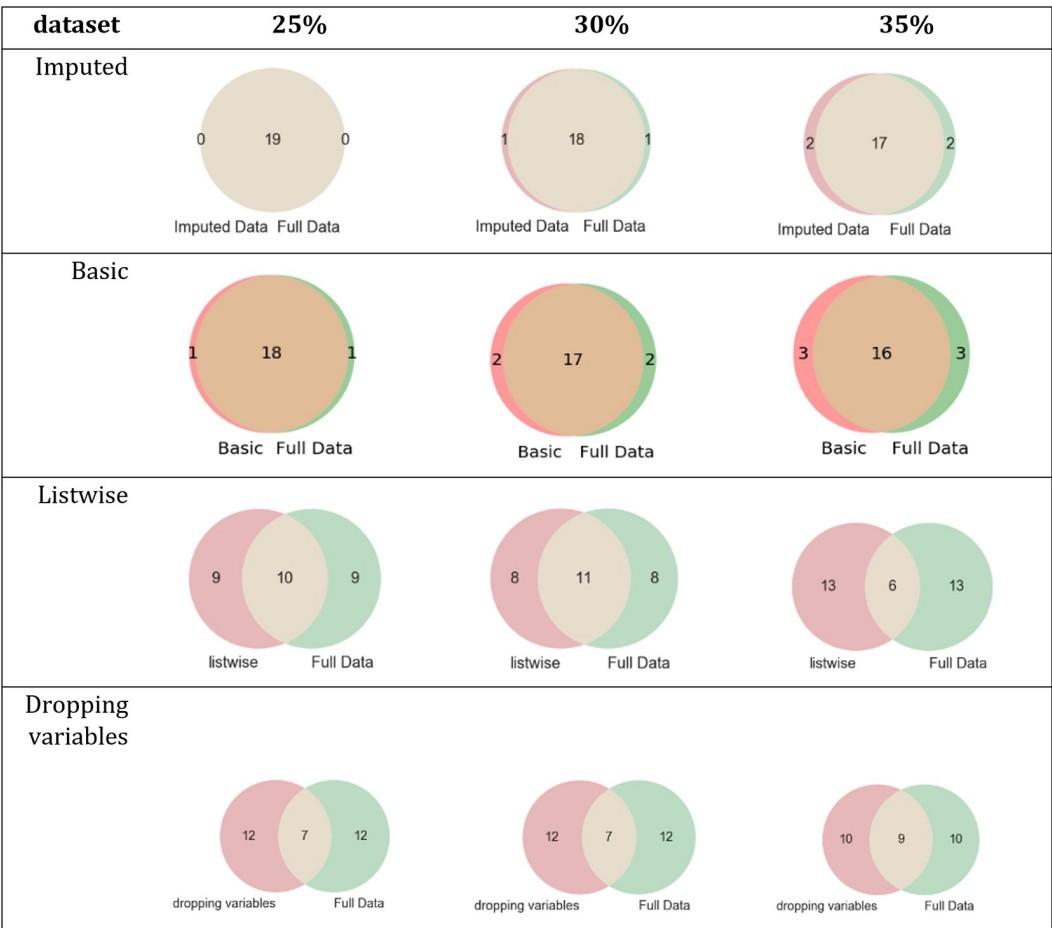

**Fig 11. Intersections of sets of relevant features of the breast-cancer dataset and its simulated datasets.**

overall accuracy achieved by MICE was better than the overall accuracy of mode replacement in 100% of missingness rates, Fig 2 and Table 2. The accuracy calculated by feature showed that some features obtained better accuracy than others, Figs 3 and 4. For feature F3, the accuracy achieved using mode replacement was better than the MICE imputations. When FS was carried out, feature F3 was not considered relevant, which meant that this feature could represent noise. Besides, analyzing the accuracies calculated for missingness rates by feature, the imputation of MICE outperformed mode replacement in 97.54% of cases.

For the *letter-recognition* dataset, the overall RMSE and MAE achieved by MICE were better than the overall RMSE and MAE of mean replacement in 100% of the overall errors, Table 3. In the analysis by the feature of missingness rates, the MAE and RMSE achieved by MICE were better than the MAE and RMSE of mean replacement in 99.62% and 96.87% of cases, respectively.

The *statlog (heart)* contained mixed numerical and categorical variables. For this dataset, the overall accuracy achieved by MICE was better than the overall accuracy of mode replacement in 100% of missingness rates analyzed, Table 4. The accuracies calculated of missingness rates by feature showed that MICE was better than the accuracy of mode replacement in 75% of cases. The overall RMSE and MAE achieved by MICE were better than the overall RMSE and MAE of mean replacement in 100% of missingness rates, Table 5. Moreover, the RMSE

and MAE calculated for missingness rates by the MICE feature outperformed the RMSE and MAE of mean replacement in 68.85% and 81.42% of cases, respectively. Features for which imputation calculated by mode/mean replacement was better than the corresponding MICE imputation were F2, F4, F5, and F6. These were not selected as relevant features in the FS process carried out on a complete *letter-recognition* dataset.

In the *spambase* dataset, Table 6, the overall RMSE and MAE achieved by MICE were better than the overall RMSE and MAE of mean replacement in 60% and 100% of missingness rates, respectively. However, in the analysis by feature, the RMSE and MAE of MICE were better than RMSE and MAE of mean replacement in 70% and 77.36%, respectively. In the results by feature, the percentages of RMSE and MAE decreased because *the spambase* dataset has a high number of features, and several of them are irrelevant and considered as noise.

In addition, Figs 5–7 show how the distribution of the *breast-cancer* dataset changed when the method of imputing data was mode replacement while the imputation performed by MICE algorithm achieved a similar distribution to the original dataset. Likewise, the *startlog* (heart), *spambase*, and *letter-recognition* datasets had changes in their distributions when the mode replacement method was employed.

For evaluating the impact of missing values in the FS process, three simulated datasets were built for each complete dataset (*breast-cancer*, *letter-recognition*, *statlog*, and *spambase*) using different missingness rate percentages (25%, 30%, and 35%) and four techniques to handle missing values were applied on each simulated dataset. The results showed the differences among the sets of relevant features of the datasets processed with techniques to handle missing values. For *letter-recognition*, the datasets imputed by MICE and complete dataset obtained the same set of relevant features, see Table 7 and Fig 8. However, the datasets imputed by basic replacement and dropped datasets changed their sets of relevant features regarding the complete dataset set of relevant features. In the *statlog* dataset, the set of relevant features of datasets imputed by MICE had two or three elements different to those of the complete dataset, see Table 8 and Fig 9. The other sets of relevant features changed in 6, 15, and 17 elements regarding the complete dataset set of relevant features. The results of FS on the *spambase* dataset showed that the most similar set to the set of relevant features of the complete dataset was the set of relevant features of the dataset imputed by MICE, Table 9 and Fig 10. For *breast-cancer*, the set of relevant features of the complete dataset and the datasets imputed by MICE differ in very few elements. The sets of relevant features of datasets imputed by basic replacement changed slightly, Table 10 and Fig 11. The sets of relevant features of datasets imputed by listwise and dropping variables have many different elements.

In general, FS results showed that the datasets imputed by using MICE obtained sets of relevant features similar to the sets of relevant features calculated using the complete datasets. Likewise, the biggest differences were found between the sets of relevant features of the complete datasets and the datasets imputed by listwise and dropping variables.

Researchers have compared methods to impute data in previous work to determine how to improve the quality of imputation or to establish which method is better for a specific mechanism of missing values, type of variables, or dataset. Nonetheless, most studies did not evaluate the impact of imputation or removing data in the feature selection process. For instance, a comparison of imputation methods was carried out in [41]. The study used a complete dataset about smoking habits to simulate datasets with missingness rates of 5% and 15%. Although the authors showed imputation results for different missingness

simulations, they only considered two missingness rates, and the dataset contained only categorical variables. Another work compared basic imputation and deletion methods. The results showed that pairwise deletion was the best technique for the dataset used in the evaluation [42]. The study evaluated missingness rates of 5%, 10%, 15%, 20% and 30%. However, the study considered neither imputation in numerical variables nor analysis of feature selection. The comparison of six methods for missing data was carried out in [43]. For the evaluation, simulated datasets were built using different missingness rate percentages (from 5% to 45%). Although the evaluation showed a detailed and reliable process to evaluate the quality of imputations calculated by the most popular methods, this did not show the impact of imputation in the feature selection process. The comparison of imputation methods in [44] also evaluated some of the most common techniques to impute data. However, the results only showed the limitations of the algorithms to impute data in any dataset. In general, most studies showed the evaluation of imputation quality but did not present the impact of missing values in subsequent analyzes. Some researchers have studied the influence of missing values in classification. However, they did not review the effect caused for missing values or imputed values in the FS process [45–48].

This study has several limitations, and the results of the quality of imputation for each method are limited to the datasets used. Hence, researchers should study their datasets to decide which method applies. In this sense, the main contribution of our research is not providing a universal solution to handle missing values or to select relevant features. Rather it involves presenting evidence about the need to consider the impact of missing values in the feature selection process.

As future work, we are considering improving the implementation of the MICE algorithm to use regression models and other methods to predict or estimate missing values. Another enhancement to ponder is evaluating whether or not the imputations improve when the target variable is included as an independent variable in predicting missing values. Besides, it is important to mention that although we designed an experiment to evaluate the impact of missing values in the feature selection process, we did not experiment simulating the three different mechanisms of missing values. For future work, we consider that the evaluation and results should be analyzed treating the mechanism of missing values separately.

## Conclusions

In this paper, the implementation and evaluation of the MICE algorithm are described. MICE was developed to handle missing data, a commonly occurring problem in real datasets. Our implementation was evaluated by calculating imputed datasets from simulated datasets with different missingness rates. The evaluation compared the imputation quality of the MICE algorithm and basic methods, and the results of feature selection on complete datasets and imputed datasets (by MICE and basic methods).

According to the overall results of accuracy, MAE and RMSE shown in the evaluation, the MICE algorithm was better than the basic methods in all missingness rates used to simulate missing values in the *breast-cancer*, *letter-recognition*, and *statlog* (heart) datasets. For the *spambase* dataset, although the MICE algorithm achieved an overall MAE in all missingness rates better than the overall MAE of the basic imputations, the RMSE of the MICE algorithm only outperformed the RMSE of the basic method in 60% of all missingness rates.

The analysis of accuracy, MAE, and RMSE by feature showed that the basic method of imputation outperformed the imputation of the MICE algorithm for some features. According

to the feature selection process applied to the complete datasets, these features were not relevant.

The evaluation results showed that for missingness rates greater than 5% and less than 50%, the complete datasets and imputed datasets calculated using MICE obtained similar distributions of their variables and similar results in the analyzes of feature selection.

Moreover, the datasets imputed using basic methods showed better results in the feature selection process than the simulated datasets handled by dropping missing variables or missing cases. However, the distribution of the variables of imputed datasets changed, meaning that the basic methods bias the datasets and accordingly that learning models could be biased.

Furthermore, selecting an appropriate method to handle missing values depends on the dataset, the mechanism of missing values, and the missingness rate. This paper showed evidence about the impact of missing values in common subsequent analyzes, such as the feature selection process.

Finally, as with any study, this work has limitations, and we cannot conclude that the MICE algorithm is the best method to handle missing values in all situations. However, the evidence presented in this paper shows that imputation could potentially be better for the avoidance of bias in subsequent analyzes than simply removing data in datasets with missing values.

## Appendixes

### Appendix A: Results of breast-cancer

**Table 11. Accuracy of MICE by feature.**

| RATE | F1 | F2 | F3 | F4 | F5 | F6 | F7 | F8 | F9 |
|------|------|------|------|------|------|------|------|------|------|
| 0.05 | 0.973 | 0.988 | 0.962 | 0.988 | 0.991 | 0.975 | 0.977 | 0.974 | 0.988 |
| 0.1 | 0.943 | 0.972 | 0.922 | 0.975 | 0.985 | 0.948 | 0.953 | 0.942 | 0.978 |
| 0.15 | 0.915 | 0.958 | 0.881 | 0.962 | 0.976 | 0.924 | 0.934 | 0.914 | 0.966 |
| 0.2 | 0.883 | 0.937 | 0.838 | 0.951 | 0.967 | 0.896 | 0.908 | 0.883 | 0.955 |
| 0.25 | 0.853 | 0.927 | 0.793 | 0.936 | 0.958 | 0.87 | 0.885 | 0.848 | 0.944 |
| 0.3 | 0.816 | 0.902 | 0.752 | 0.926 | 0.951 | 0.843 | 0.862 | 0.818 | 0.934 |
| 0.35 | 0.795 | 0.884 | **0.712** | 0.91 | 0.938 | 0.816 | 0.836 | 0.786 | 0.921 |
| 0.4 | 0.758 | 0.861 | **0.668** | 0.9 | 0.93 | 0.794 | 0.809 | 0.759 | 0.911 |
| 0.45 | 0.722 | 0.834 | 0.63 | 0.887 | 0.917 | 0.763 | 0.791 | 0.725 | 0.899 |

**Table 12. Accuracy of Mode by feature.**

| RATE | F1 | F2 | F3 | F4 | F5 | F6 | F7 | F8 | F9 |
|------|------|------|------|------|------|------|------|------|------|
| 0.05 | 0.962 | 0.978 | 0.96 | 0.988 | 0.99 | 0.973 | 0.977 | 0.969 | 0.988 |
| 0.1 | 0.929 | 0.954 | 0.919 | 0.973 | 0.981 | 0.945 | 0.953 | 0.936 | 0.978 |
| 0.15 | 0.891 | 0.93 | 0.875 | 0.961 | 0.971 | 0.92 | 0.929 | 0.904 | 0.966 |
| 0.2 | 0.859 | 0.906 | 0.835 | 0.949 | 0.96 | 0.89 | 0.903 | 0.874 | 0.953 |
| 0.25 | 0.821 | 0.882 | 0.791 | 0.933 | 0.949 | 0.86 | 0.875 | 0.839 | 0.942 |
| 0.3 | 0.791 | 0.859 | 0.752 | 0.924 | 0.941 | 0.831 | 0.85 | 0.808 | 0.931 |
| 0.35 | 0.758 | 0.837 | **0.714** | 0.909 | 0.932 | 0.803 | 0.828 | 0.779 | 0.919 |
| 0.4 | 0.722 | 0.815 | **0.669** | 0.899 | 0.922 | 0.776 | 0.8 | 0.748 | 0.906 |
| 0.45 | 0.692 | 0.794 | 0.626 | 0.888 | 0.911 | 0.744 | 0.775 | 0.711 | 0.893 |

## Appendix B: Results of statlog (heart)

**Table 13. Accuracy of MICE by feature.**

| RATE | F2 | F3 | F6 | F7 | F9 | F13 |
|------|------|------|------|------|------|------|
| 0.05 | 0.984 | 0.978 | 0.991 | 0.977 | 0.988 | 0.986 |
| 0.1 | **0.967** | 0.952 | 0.982 | 0.954 | 0.973 | 0.967 |
| 0.15 | **0.951** | 0.928 | 0.976 | 0.931 | 0.961 | 0.949 |
| 0.2 | 0.932 | 0.903 | **0.968** | 0.907 | 0.943 | 0.931 |
| 0.25 | **0.916** | 0.878 | **0.961** | 0.884 | 0.93 | 0.914 |
| 0.3 | **0.901** | 0.854 | **0.954** | 0.855 | 0.91 | 0.897 |
| 0.35 | **0.879** | 0.828 | **0.947** | 0.833 | 0.895 | 0.881 |
| 0.4 | **0.867** | 0.803 | **0.939** | 0.801 | 0.878 | 0.858 |
| 0.45 | **0.849** | 0.772 | **0.93** | 0.78 | 0.863 | 0.837 |
| 0.5 | **0.833** | 0.75 | **0.922** | 0.756 | 0.843 | 0.813 |

**Table 14. Accuracy of mode replacement by feature.**

| RATE | F2 | F3 | F6 | F7 | F9 | F13 |
|------|------|------|------|------|------|------|
| 0.05 | 0.984 | 0.976 | 0.991 | 0.974 | 0.985 | 0.98 |
| 0.1 | **0.968** | 0.95 | 0.982 | 0.946 | 0.967 | 0.957 |
| 0.15 | **0.952** | 0.924 | 0.976 | 0.922 | 0.951 | 0.935 |
| 0.2 | 0.932 | 0.897 | **0.969** | 0.895 | 0.934 | 0.912 |
| 0.25 | **0.918** | 0.872 | **0.962** | 0.865 | 0.921 | 0.888 |
| 0.3 | **0.902** | 0.848 | **0.956** | 0.838 | 0.902 | 0.863 |
| 0.35 | **0.885** | 0.82 | **0.95** | 0.81 | 0.886 | 0.848 |
| 0.4 | **0.871** | 0.797 | **0.942** | 0.788 | 0.867 | 0.821 |
| 0.45 | **0.856** | 0.771 | **0.934** | 0.764 | 0.851 | 0.802 |
| 0.5 | **0.841** | 0.748 | **0.926** | 0.741 | 0.837 | 0.78 |

**Table 15. MAE of MICE by feature.**

| RATE | F1 | F4 | F5 | F8 | F10 | F11 | F12 |
|------|------|------|------|------|------|------|------|
| 0.05 | 0.133 | 0.123 | 0.089 | 0.112 | 0.12 | 0.199 | 0.212 |
| 0.1 | 0.134 | 0.125 | 0.09 | 0.119 | 0.112 | 0.195 | 0.22 |
| 0.15 | 0.133 | 0.128 | 0.088 | 0.12 | 0.119 | 0.203 | 0.223 |
| 0.2 | 0.136 | 0.129 | **0.09** | 0.12 | 0.118 | 0.202 | 0.225 |
| 0.25 | 0.144 | **0.132** | **0.093** | 0.125 | 0.119 | 0.214 | 0.236 |
| 0.3 | 0.145 | **0.14** | **0.094** | 0.13 | 0.125 | 0.215 | 0.242 |
| 0.35 | 0.154 | **0.147** | **0.097** | 0.14 | 0.129 | 0.225 | 0.242 |
| 0.4 | 0.157 | **0.151** | **0.102** | 0.143 | 0.132 | 0.233 | 0.251 |
| 0.45 | 0.157 | **0.15** | **0.105** | 0.141 | 0.133 | 0.236 | 0.253 |
| 0.5 | 0.156 | **0.148** | **0.1** | 0.141 | 0.134 | 0.241 | 0.253 |

**Table 16. MAE of mode replacement by feature.**

| RATE | F1 | F4 | F5 | F8 | F10 | F11 | F12 |
|------|------|------|------|------|------|------|------|
| 0.05 | 0.165 | 0.126 | 0.091 | 0.134 | 0.158 | 0.286 | 0.264 |
| 0.1 | 0.159 | 0.129 | 0.094 | 0.143 | 0.147 | 0.28 | 0.26 |
| 0.15 | 0.156 | 0.13 | 0.09 | 0.143 | 0.149 | 0.282 | 0.267 |
| 0.2 | 0.156 | 0.131 | **0.089** | 0.142 | 0.147 | 0.282 | 0.268 |
| 0.25 | 0.159 | **0.13** | **0.089** | 0.143 | 0.146 | 0.284 | 0.271 |
| 0.3 | 0.156 | **0.13** | **0.087** | 0.144 | 0.149 | 0.281 | 0.27 |
| 0.35 | 0.159 | **0.131** | **0.089** | 0.144 | 0.147 | 0.281 | 0.267 |
| 0.4 | 0.157 | **0.13** | **0.088** | 0.146 | 0.147 | 0.283 | 0.267 |
| 0.45 | 0.157 | **0.13** | **0.088** | 0.145 | 0.148 | 0.283 | 0.266 |
| 0.5 | 0.157 | **0.131** | **0.087** | 0.144 | 0.147 | 0.283 | 0.268 |

**Table 17. RMSE of MICE by feature.**

| RATE | F1 | F4 | F5 | F8 | F10 | F11 | F12 |
|------|------|------|------|------|------|------|------|
| 0.05 | 0.163 | 0.154 | 0.115 | 0.138 | 0.155 | 0.25 | 0.272 |
| 0.1 | 0.166 | 0.16 | 0.123 | 0.149 | 0.147 | 0.245 | 0.283 |
| 0.15 | 0.165 | 0.163 | 0.12 | 0.149 | 0.157 | 0.256 | 0.289 |
| 0.2 | 0.166 | 0.165 | **0.121** | 0.152 | 0.156 | 0.256 | 0.298 |
| 0.25 | 0.178 | **0.169** | **0.121** | 0.157 | 0.159 | 0.27 | 0.31 |
| 0.3 | 0.179 | **0.179** | **0.124** | 0.164 | 0.167 | 0.274 | 0.321 |
| 0.35 | 0.19 | **0.187** | **0.128** | 0.178 | 0.173 | 0.286 | **0.323** |
| 0.4 | **0.195** | 0.193 | **0.135** | **0.184** | 0.175 | 0.294 | **0.334** |
| 0.45 | **0.194** | 0.191 | **0.139** | 0.18 | 0.177 | 0.297 | **0.334** |
| 0.5 | **0.195** | 0.189 | **0.132** | **0.179** | 0.176 | 0.299 | **0.33** |

**Table 18. RMSE of mode replacement by feature.**

| RATE | F1 | F4 | F5 | F8 | F10 | F11 | F12 |
|------|------|------|------|------|------|------|------|
| 0.05 | 0.197 | 0.157 | 0.116 | 0.163 | 0.191 | 0.31 | 0.307 |
| 0.1 | 0.191 | 0.163 | 0.125 | 0.177 | 0.179 | 0.301 | 0.305 |
| 0.15 | 0.189 | 0.166 | 0.121 | 0.174 | 0.186 | 0.307 | 0.314 |
| 0.2 | 0.189 | 0.166 | **0.12** | 0.174 | 0.183 | 0.306 | 0.317 |
| 0.25 | 0.191 | **0.164** | **0.117** | 0.176 | 0.182 | 0.309 | 0.322 |
| 0.3 | 0.188 | **0.167** | **0.116** | 0.176 | 0.187 | 0.304 | 0.321 |
| 0.35 | 0.191 | **0.168** | **0.117** | 0.178 | 0.183 | 0.305 | **0.316** |
| 0.4 | **0.189** | **0.166** | **0.116** | **0.179** | 0.184 | 0.309 | **0.317** |
| 0.45 | **0.189** | **0.168** | **0.116** | 0.18 | 0.187 | 0.309 | **0.316** |
| 0.5 | **0.189** | **0.168** | **0.114** | **0.178** | 0.184 | 0.309 | **0.32** |

## Appendix C: Results of feature selection on letter-recognition

**Table 19. Results of feature selection of simulated datasets.**

| dataset | Algorithm | 25% | 30% | 35% |
|---|---|---|---|---|
| Imputed | Select K Best (Chi-squared) | F11, F13, F15 | F11, F13, F15 | F11, F13, F15 |
| | Select K Best (F-value) | F7, F11, F14 | F7, F11, F14 | F7, F11, F14 |
| | Select K Best (ANOVA F-value) | F7, F11, F13 | F7, F11, F13 | F7, F11, F13 |
| | Feature Recursive Elimination | F12, F13, F14 | F12, F13, F14 | F12, F13, F14 |
| | Feature Importance | F9, F13, F15 | F9, F13, F15 | F9, F13, F15 |
| Basic | Select K Best (Chi-squared) | **F8, F9**, F13 | **F8, F9**, F13 | **F9**, F13, F15 |
| | Select K Best (F-value) | F7, F11, F14 | F7, F11, F14 | F7, F11, F14 |
| | Select K Best (ANOVA F-value) | **F2, F13**, F14 | **F2, F13**, F14 | **F2, F13**, F14 |
| | Feature Recursive Elimination | F12, F13, F14 | F12, F13, F14 | F12, F13, F14 |
| | Feature Importance | **F8**, F9, F13 | **F8**, F9, **F13** | F9, **F12, F14** |
| Listwise | Select K Best (Chi-squared) | **F8**, F13, F15 | **F2**, F13, F15 | **F5**, F13, F15 |
| | Select K Best (F-value) | F7, F11, F14 | **F9**, F11, F14 | F7, F11, F14 |
| | Select K Best (ANOVA F-value) | F7, F11, **F12** | F7, F11, F13 | F11, **F12, F14** |
| | Feature Recursive Elimination | **F3, F5**, F13 | F12, F14, **F15** | F12, F13, F14 |
| | Feature Importance | F9, **F12**, F13 | F9, F13, F15 | **F12**, F13, **F16** |
| Dropping variables | Select K Best (Chi-squared) | **F6, F7, F10** | **F7, F8**, F11 | **F6, F7, F9** |
| | Select K Best (F-value) | **F9**, F11, **F12** | F7, **F10, F12** | **F8, F10**, F11 |
| | Select K Best (ANOVA F-value) | **F9, F10**, F11 | **F10**, F11, **F12** | **F8, F9, F10** |
| | Feature Recursive Elimination | **F9, F10, F11** | **F10, F11, F12** | **F8, F9, F10** |
| | Feature Importance | **F6, F7, F10** | **F7, F8**, F11 | **F6, F7, F9** |

## Appendix D: Result of feature selection on statlog(heart)

**Table 20. Results of feature selection of simulated datasets.**

| dataset | Algorithm | 25% | 30% | 35% |
|---|---|---|---|---|
| **Imputed** | Select K Best (Chi-squared) | F3, F9, F12, F13 | F3, F9, F12, F13 | F3, F9, F12, F13 |
| | Select K Best (F-value) | F3, **F10**, F12, F13 | F9, **F10**,F12, F13 | F3, F9, **F10**, F13 |
| | Select K Best (ANOVA F-value) | F3, **F10**, F12, F13 | F9, **F10**,F12, F13 | F3, F9, **F10**, F13 |
| | Feature Recursive Elimination | F8, F10, F12 | F8, F10, F12 | F8, F10, F12 |
| | Feature Importance | F3, F9, F12, F13 | F3, **F10**, F12, F13 | F3, F9, F12, F13 |
| **Basic** | Select K Best (Chi-squared) | F3,**F10**,F12, F13 | F3,**F10**,F12, F13 | F3,**F10**,F12, F13 |
| | Select K Best (F-value) | **F8,F10**,F12, F13 | F3, **F8,F10**,F12 | **F8,F10**,F12, F13 |
| | Select K Best (ANOVA F-value) | **F8,F10**,F12, F13 | F3, **F8,F10**,F12 | **F8,F10**,F12, F13 |
| | Feature Recursive Elimination | F8, F10, F12 | F8, F10, F12 | F8, F10, F12 |
| | Feature Importance | F3, **F10**,F12, F13 | F3, **F8,F10**,F12 | **F8,F10**,F12, F13 |
| **listwise** | Select K Best (Chi-squared) | **F2**, F3, **F11**, F12 | **F2**, F3, F9, F13 | **F2, F7, F8**, F9 |
| | Select K Best (F-value) | **F2, F5, F7**, F12 | **F4, F5, F8**, F12 | **F2, F6, F7**, F9 |
| | Select K Best (ANOVA F-value) | **F2, F5, F7**, F12 | F3, **F8**, F9, F13 | **F2, F6, F7**, F9 |
| | Feature Recursive Elimination | **F2, F3, F11** | **F3, F7, F13** | **F6, F7, F9** |
| | Feature Importance | **F2, F3, F5, F7** | F3, F8, **F9**, F13 | **F2, F6, F7**, F9 |
| **Dropping variables** | Select K Best (Chi-squared) | **F2, F8**, F9, **F10** | **F2, F8**, F9, **F10** | F3, **F7, F8**, F9 |
| | Select K Best (F-value) | **F7, F8**, F9, **F10** | F3, **F7, F8, F10** | F3, **F6, F7**, F9 |
| | Select K Best (ANOVA F-value) | **F7, F8**, F9, **F10** | F3, **F7, F8, F10** | F3, **F6, F7**, F9 |
| | Feature Recursive Elimination | **F4, F7, F8** | **F7, F8**, F10 | **F6, F7, F9** |
| | Feature Importance | **F3, F7, F8, F10** | **F3, F7, F8, F10** | F3, **F6, F7**, F9 |

## Appendix E: Result of feature selection on spambase

**Table 21. Results of feature selection of simulated datasets.**

| dataset | Algorithm | 25% | 30% | 35% |
|---|---|---|---|---|
| **Imputed** | SKB (Chi-squared) | F25, F27, F55, F56, F57 | F25, F27, F55, F56, F57 | F25, F27, F55, F56, F57 |
| | SKB (F-value) | F7, F19, F21, F23, F53 | F7, F19, F21, F23, F53 | F7, **F17**, F21, F23, **F56** |
| | SKB (ANOVA F-value) | F7, F19, F21, F23, F53 | F7, F19, F21, F23, F53 | F7, **F17**, F21, F23, **F56** |
| | FRE | F7, **F23**, F53 | F7, **F23**, F53 | F7, **F41**, F53 |
| | Feature Importance | F7, F16, F21, F52, F53 | F7, F16, F21, F52, F53 | F7, F21, F52, F53, **F56** |
| **Basic** | SKB (Chi-squared) | F25, F27, F55, F56, F57 | **F16**, F27, F55, F56, F57 | **F16**, F27, F55, F56, F57 |
| | SKB (F-value) | F7, **F16**, F21, F23, F53 | **F7, F16**, F21, F23, F53 | F7, **F16, F17**, F21, F53 |
| | SKB (ANOVA F-value) | F7, **F16**, F21, F23, F53 | **F7, F16**, F21, F23, F53 | F7, **F16, F17**, F21, F53 |
| | FRE | F7, **F23**, F53 | F7, **F23**, F53 | F7, **F23**, F53 |
| | Feature Importance | F7, F16, F21, F52, F53 | F16, F21, **F23**, F52, F53 | F16, F21, F52, **F56, F55** |
| **listwise** | SKB (Chi-squared) | **F16**, F27, F55, F56, F57 | F25, F27, F55, F56, F57 | **F22**, F27, F55, F56, F57 |
| | SKB (F-value) | F7, **F16**, F21, F23, **F57** | **F16**, F21, F23, F53, **F56** | **F8, F17**, F21, **F52**, F53 |
| | SKB (ANOVA F-value) | F7, **F16**, F21, F23, **F57** | **F16**, F21, F23, F53, **F56** | **F8, F17**, F21, **F52**, F53 |
| | FRE | F7, F24, F53 | **F16, F23**, F53 | **F16, F21**, F27 |
| | Feature Importance | F7, F16, F21, F52, F53 | **F5**, F16, F21, F52, F53 | F16, **F17**, F21, F52, **F56** |
| **Dropping variables** | SKB (Chi-squared) | **F18, F20, F21, F42, F43** | **F12, F16, F38, F39, F40** | **F9, F12, F34, F37, F38** |
| | SKB (F-value) | **F14, F16, F18, F19, F40** | **F6, F12, F13, F16, F18** | **F9, F10, F12, F14, F35** |
| | SKB (ANOVA F-value) | **F14, F16, F18, F19, F40** | **F6, F12, F13, F16, F18** | **F9, F10, F12, F14, F35** |
| | FRE | **F19, F21, F40** | **F16, F18, F25** | **F14, F19, F35** |
| | Feature Importance | **F13, F18, F39, F40, F42** | F6, **F12**, F16, **F18, F37** | **F9, F12, F34, F35, F37** |

## Appendix F: Result of feature selection breast-cancer

**Table 22. Results of feature selection of simulated datasets.**

| dataset | Algorithm | 25% | 30% | 35% |
|---|---|---|---|---|
| **Imputed** | Select K Best (Chi-squared) | F3, F4, F5, F6 | F3, F4, F5, F6 | F3, F4, F5, F6 |
| | Select K Best (F-value) | F4, F5, F6, F9 | F4, F5, F6, F9 | F4, F5, F6, F9 |
| | Select K Best (ANOVA F-value) | F4, F5, F6, F9 | F4, F5, F6, F9 | F4, F5, F6, F9 |
| | Feature Recursive Elimination | F5, F6, F7 | **F1**, F5, F6 | **F1**, F5, F6 |
| | Feature Importance | F1, F3, F6, F8 | F1, F3, F6, F8 | F1, F3, **F4**, F8 |
| **Basic** | Select K Best (Chi-squared) | F3, F4, F5, F6 | F3, F4, F5, F6 | F3, F4, F6, **F9** |
| | Select K Best (F-value) | F4, F5, F6, F9 | F4, F5, F6, F9 | F4, F5, F6, F9 |
| | Select K Best (ANOVA F-value) | F4, F5, F6, F9 | F4, F5, F6, F9 | F4, F5, F6, F9 |
| | Feature Recursive Elimination | **F1**, F6, F7 | **F1, F4**, F6 | F5, F7, **F9** |
| | Feature Importance | F1, F3, F6, F8 | F1, F3, F6, F8 | F1, F3, **F4**, F8 |
| **listwise** | Select K Best (Chi-squared) | **F1, F2**, F3, F4 | F3, F4, F5, **F8** | **F1, F2**, F4, **F7** |
| | Select K Best (F-value) | **F3**, F5, **F7**, F9 | **F3**, F4, F5, F6 | **F1, F2**, F4, **F7** |
| | Select K Best (ANOVA F-value) | **F3**, F5, **F7**, F9 | **F3**, F4, F5, F6 | **F1, F2**, F4, **F7** |
| | Feature Recursive Elimination | F5, F7, **F9** | **F2, F4**, F7 | **F2**, F7, **F8** |
| | Feature Importance | F1, F3, **F4**, **F7** | **F4, F5**, F6, **F7** | **F2**, F3, **F7**, F8 |
| **pairwise** | Select K Best (Chi-squared) | **F2, F3**, F4, **F7** | **F2**, F3, F4, **F7** | **F1**, F3, F4, F6 |
| | Select K Best (F-value) | **F2, F3**, F4, **F7** | **F2, F3**, F4, **F7** | **F1, F3**, F4, F6 |
| | Select K Best (ANOVA F-value) | **F2, F3**, F4, **F7** | **F2, F3**, F4, **F7** | **F1, F3**, F4, F6 |
| | Feature Recursive Elimination | **F4**, F5, F7 | **F3, F4**, F5 | **F1, F3, F4** |
| | Feature Importance | **F2**, F3, **F4**, F6 | F1, **F2, F4**, F6 | F1, F3, **F4, F5** |

## Acknowledgments

The authors are grateful too to Mr Colin McLachlan for suggestions related to the English text.

## Author Contributions

**Conceptualization:** Maritza Mera-Gaona, Ursula Neumann.

**Formal analysis:** Maritza Mera-Gaona, Ursula Neumann.

**Investigation:** Maritza Mera-Gaona, Ursula Neumann.

**Methodology:** Maritza Mera-Gaona.

**Software:** Maritza Mera-Gaona.

**Supervision:** Ursula Neumann, Rubiel Vargas-Canas, Diego M. López.

**Validation:** Maritza Mera-Gaona, Ursula Neumann.

**Writing – original draft:** Maritza Mera-Gaona.

**Writing – review & editing:** Ursula Neumann, Rubiel Vargas-Canas, Diego M. López.

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
