## [Decision Letter · Decision Letter 0]

15 Mar 2021

PONE-D-20-39195

Evaluating the impact of Multivariate Imputation by MICE in Feature Selection

PLOS ONE

Dear Dr. Mera-Gaona,

Thank you for submitting your manuscript to PLOS ONE. After careful consideration, we feel that it has merit but does not fully meet PLOS ONE’s publication criteria as it currently stands. Therefore, we invite you to submit a revised version of the manuscript that addresses the points raised during the review process.

We look forward to receiving your revised manuscript.

Kind regards,

Zaher Mundher Yaseen

Academic Editor

PLOS ONE

Journal Requirements:

We note that one or more of the authors are employed by a commercial company: FraunhoferCenter for Applied Research on Supply Chain Services SCS, Nuremberg, Germany

(2) Please also provide an updated Competing Interests Statement declaring this commercial affiliation along with any other relevant declarations relating to employment, consultancy, patents, products in development, or marketed products, etc.  

Reviewers' comments:

Reviewer's Responses to Questions

**Comments to the Author**

1. Is the manuscript technically sound, and do the data support the conclusions?

Reviewer #1: Partly

Reviewer #2: Partly

2. Has the statistical analysis been performed appropriately and rigorously? 

Reviewer #1: Yes

Reviewer #2: Yes

3. Have the authors made all data underlying the findings in their manuscript fully available?

Reviewer #1: Yes

Reviewer #2: No

4. Is the manuscript presented in an intelligible fashion and written in standard English?

Reviewer #1: Yes

Reviewer #2: No

5. Review Comments to the Author

Reviewer #1: This paper provides the experiments to demonstrate the positive impact of MICE in the feature selection methods to handle data sets with missing values. The experimental results show that feature selection with MICE can achieve better performance compared with traditional imputation methods.

Here are some concerns:

1. This paper should provide a more comprehensive survey to introduce some related work, including feature selection and imputation methods.

2. This paper only uses 4 data sets. It would be better to use more data sets. Moreover, since feature selection is often used to handle high-dimensional data sets, it would be better to use some high-dimensional data sets. Note that, the dimensions of used data are too low (i.e., they only contain several tens of features).

3. It would be better to conduct experiments with some state-of-the-art feature selection methods, including some filter, embedding and wrapper feature selection methods.

4. The paper needs a more careful proofreading.

Reviewer #2: The topic is important. The results can be useful. However, there are a number of issues that require attention. These are listed below. Note that some research papers are mentioned below which may be consulted and cited if the authors wish, Or better quality research papers may be used and cited instead):

(1)The literature review of this paper is not satisfactory. It does not clearly describe the relevant research work. A strong literature review should be done. Multivariate imputation by chained equations (MICE) is an existing method described in the literature. The references should be cited at the beginning. In the review, feature selection in relevant context also needs to be discussed. A number of papers have described how the important features are extracted from a dataset, some of the works are: (a) Data-Driven Diagnosis of Spinal Abnormalities Using Feature Selection and Machine Learning Algorithms. PLOS One, 2020. (b) Prediction of Chronic Kidney Disease - A Machine Learning Perspective," in IEEE Access, 2021. (c) An Optimized Stacked Support Vector Machines Based Expert System for the Effective Prediction of Heart Failure, IEEE Access, 2019.

(2)The motivation of this research must be clearly and elaborately described. Why this research is important? Currently brief discussion is provided which is not enough. Also Section 4 (discussion) gives some indication, but that is at the end of the paper. So, some of that discussion can be placed at the literature review section.

(3)In Section 3.1.1.3 and in a number of places “100%” is used. Such as “MICE's overall accuracy was 100% better than the overall accuracy of mode replacement”. It is not clear why this is used. Please clarify how this wording makes sense.

(4)In Section 3 – the results section, the authors present a number of graphs and results. However, there is little explanation of the results.

(5)In Section 3 – the results section, a formatting issue is present in the form of “Error! Reference source not found”. This needs to be corrected.

(6)Overall the presentation and writing of the paper need some editing. Currently the results and findings are not easy to understand. Particularly, the concept “the impact of imputation in the feature selection process” is not clearly understood in the paper. Also a clearer discussion is required for the “the use of multiple datasets of different diseases and the difference in findings based on datasets”.

6. PLOS authors have the option to publish the peer review history of their article (what does this mean?). If published, this will include your full peer review and any attached files.

Reviewer #1: No

Reviewer #2: No

---

## [Author Response · Author response to Decision Letter 0]

25 May 2021

Manuscript ID: PONE-D-20-39195

Title: Evaluating the impact of multivariate imputation by MICE in feature selection

Dear Reviewers,

We have uploaded a revised version of the paper. We appreciate the valuable comments, which had contributed to improving the paper’s quality.

A detailed description of changes considering all issues mentioned in your comments, and an explanation of every change made, point by point, are appended below.

Comments to the Author

3. Have the authors made all data underlying the findings in their manuscript fully available?

Response: We used 4 public datasets available in the UCI Machine Learning dataset repository. The description of the datasets used was included in the section “Materials and Methods”.

The 4 datasets are available in references [24], [25], [26] [27]

Reviewer #1: Yes

Reviewer #2: No

4. Is the manuscript presented in an intelligible fashion and written in standard English?

Reviewer #1: Yes

Reviewer #2: No

Response: Thank you for the comments. We have used a local proofreading service following your recommendation. The information of the reviewer was included in the Acknowledgments section.

Response to Reviewer 1 Comments

Reviewer #1: This paper provides the experiments to demonstrate the positive impact of MICE in the feature selection methods to handle data sets with missing values. The experimental results show that feature selection with MICE can achieve better performance compared with traditional imputation methods.

Here are some concerns:

1. This paper should provide a more comprehensive survey to introduce some related work, including feature selection and imputation methods.

Response: We appreciate your suggestions and recommendations. 

We have extended the introduction section, to better introduce the related work. We consider that, in the previous version, we did not describe well the motivation of our research and this could cause a misunderstanding of the contributions of our work. We find out there are many imputation methods and FS algorithms in the literature, however, previous studies (Feature Selection and Imputation methods) did not show how the imputation methods could bias the datasets and the Feature Selection processes. 

Besides, Most Feature Selection algorithms are designed for datasets where complete data and imputation methods are evaluated to show the quality of the imputation. However, they do not show how the imputation impacts the subsequent analyzes (feature selection, or classification).

We added a description of the protocol followed to explore the state of the art in data imputation and feature selection: the result of these reviews allows us to add references of related works in both fields. However, please, consider that the objective of our manuscript is not to deepen in the advances of Feature Selection or Data Imputation. Instead, we wanted to demonstrate in the introduction section that there are many studies in these areas, but they do not explore the effect of Imputation data in the Feature selection process. Subsequently, the comparison of our work with the related works is introduced in the Discussion section.

We added the following lines in the Introduction materials and methods Section to show how we can carried out the systematic mapping:

“To review works related to FS and data imputation, we carried out two systematic mappings focused on identifying studies related to imputation and the assembly of feature selection algorithms following the guidelines described by Petersen [5]. We used two search strings, one for each topic: 

Imputation data: (imputation data) and (missing values or missingness rates or incomplete data or incomplete dataset)

Feature selection: ("framework" and "ensemble") and ("dimensionality reduction" or "feature selection") and ("EEG" and "automatic") and ("detector" or "reading" or "recognition" or “analysis”).

The searches guided by the previous keywords, were used to find relevant papers from IEEE, PubMed, and Science Direct databases. The analysis of the papers was led following review criteria based on the quality of their contributions, particularly the proposal of imputation and assembly of feature selection algorithms ”…

Also the following sentences were added to clarify the manuscripts contributions:

… “However, when a dataset has missing values in the features, we must find a way to handle the missing values and perform preprocessing tasks to get a dataset with complete data. Commonly, the missing data problem is solved by removing the instances or features with missing values or replacing the missing values using basic mechanisms such as mean, mode, etc. Although these strategies are easy to implement, they change the distribution of the datasets and may bias subsequent Machine Learning analyzes, for instance, the feature selection or classification processes. On one hand, the methods used to handle missing values could eliminate from the dataset: (i) relevant features or (ii) instances that reveal the importance of the relevant features. On the other hand, the machine learning models could be trained using only a part of the original data points.”…

…” 

In previous studies, we evaluated how feature selection improved the performance of the classification of epileptic events and normal brain activity in Electroencephalograms [18][19]. The experiments were carried out using datasets with high dimensionality in a scenario with the need of reducing the computational complexity. The results indicated that the best subset of relevant features was selected by an approach based on Ensemble Feature Selection (EFS). 

We thus proposed a Framework of Ensemble Feature Selection to improve the selection of relevant features in datasets with high dimensionality [20]. Nonetheless, one of the weaknesses of the original proposal was the handling of datasets with missing values. In the real world, datasets have a high probability of having incomplete data, which means that handling missing values is necessary before selecting relevant features. This renders the results of FS uncertain when the dataset has incomplete data.

”..

2. This paper only uses 4 data sets. It would be better to use more data sets. Moreover, since feature selection is often used to handle high-dimensional data sets, it would be better to use some high-dimensional data sets. Note that, the dimensions of used data are too low (i.e., they only contain several tens of features).

Response: We appreciate your suggestions and recommendations.

The reason why we selected these 4 well-known datasets was that these datasets not only have been used in previous studies of FS but also because our goal was to evidence how the handling of missing data methods change the distribution of the datasets and how this affects the FS process. In this study, handling “too low” dimensionality allowed us to show graphically the effect of the changes. Even, this task was more difficult to perform for the spambase dataset, which has 57 features. 

3. It would be better to conduct experiments with some state-of-the-art feature selection methods, including some filter, embedding, and wrapper feature selection methods.

Response: thank you for your comment. 

We described in the evaluation section, the results of experiments with (a) three filter algorithms based on three statistical methods (Chi-squared, F-test, & ANOVA-test), (b) a wrapper method: Feature Importance measures for tree models, and (c) an embedded method: Recursive Feature Elimination. 

We kindly ask you to consider that we want to show how the imputation methods can help us to stabilize the results of the FS process in scenarios where we must deal with missing data. This is the main reason why we generated datasets with different rates of missing data, imputed data in the generated datasets, then, we applied feature selection algorithms, and finally we compared the relevant features selected in imputed datasets with the relevant features selected in the original datasets. 

4. The paper needs a more careful proofreading.

Response: Thank you for the comments. We have used a local proofreading service following your recommendation. 

Response to Reviewer 2 Comments

Reviewer #2: The topic is important. The results can be useful. However, there are a number of issues that require attention. These are listed below. Note that some research papers are mentioned below which may be consulted and cited if the authors wish, Or better quality research papers may be used and cited instead):

(1)The literature review of this paper is not satisfactory. It does not clearly describe the relevant research work. A strong literature review should be done. Multivariate imputation by chained equations (MICE) is an existing method described in the literature. The references should be cited at the beginning. In the review, feature selection in a relevant context also needs to be discussed. Several papers have described how the important features are extracted from a dataset, some of the works are (a) Data-Driven Diagnosis of Spinal Abnormalities Using Feature Selection and Machine Learning Algorithms. PLOS One, 2020. (b) Prediction of Chronic Kidney Disease - A Machine Learning Perspective," in IEEE Access, 2021. (c) An Optimized Stacked Support Vector Machines Based Expert System for the Effective Prediction of Heart Failure, IEEE Access, 2019.

Response: We appreciate your suggestions. 

In addition to the studies found in the systematic mappings carried out, we have included the references mentioned by the reviewer. 

Furthermore, we have extended the introduction section, to better introduce the related work. We consider that we did not describe well the motivation of our research and this could cause a misunderstanding of the contributions of our work. We find out there are many imputation methods and FS algorithms in the literature, however, previous studies (Feature Selection and Imputation methods) did not show how the imputation methods could bias the datasets and the Feature Selection processes. 

Besides, Most Feature Selection algorithms are designed for datasets where complete data and imputation methods are evaluated to show the quality of the imputation. However, they do not show how the imputation impacts the subsequent analyzes (feature selection, or classification).

We added a description of the protocol followed to explore the state of the art in data imputation and feature selection: the result of these reviews allowed us to add references of related works in both fields. However, please, consider that the objective of our manuscript is not to deepen in the advances of Feature Selection or Data Imputation. Instead, we wanted to demonstrate in the introduction section that there are many studies in these areas, but they do not explore the effect of Imputation data in the Feature selection process. Subsequently, the comparison of our work with the related works is introduced in the Discussion section.

Besides, We added the following lines in the materials and methods Section to show how we can carried out the systematic mapping

“To review works related to FS and data imputation, we carried out two systematic mappings focused on identifying studies related to imputation and the assembly of feature selection algorithms following the guidelines described by Petersen [5]. We used two search strings, one for each topic: 

Imputation data: (imputation data) and (missing values or missingness rates or incomplete data or incomplete dataset)

Feature selection: ("framework" and "ensemble") and ("dimensionality reduction" or "feature selection") and ("EEG" and "automatic") and ("detector" or "reading" or "recognition" or “analysis”).

The searches guided by the previous keywords, were used to find relevant papers from IEEE, PubMed, and Science Direct databases. The analysis of the papers was led following review criteria based on the quality of their contributions, particularly the proposal of imputation and assembly of feature selection algorithms ”…

Also the following sentences were added to clarify the manuscripts contributions:

… “However, when a dataset has missing values in the features, we must find a way to handle the missing values and perform preprocessing tasks to get a dataset with complete data. Commonly, the missing data problem is solved by removing the instances or features with missing values or replacing the missing values using basic mechanisms such as mean, mode, etc. Although these strategies are easy to implement, they change the distribution of the datasets and may bias subsequent Machine Learning analyzes, for instance, the feature selection or classification processes. On one hand, the methods used to handle missing values could eliminate from the dataset: (i) relevant features or (ii) instances that reveal the importance of the relevant features. On the other hand, the machine learning models could be trained using only a part of the original data points.”…

…” 

In previous studies, we evaluated how feature selection improved the performance of the classification of epileptic events and normal brain activity in Electroencephalograms [18][19]. The experiments were carried out using datasets with high dimensionality in a scenario with the need of reducing the computational complexity. The results indicated that the best subset of relevant features was selected by an approach based on Ensemble Feature Selection (EFS). 

We thus proposed a Framework of Ensemble Feature Selection to improve the selection of relevant features in datasets with high dimensionality [20]. Nonetheless, one of the weaknesses of the original proposal was the handling of datasets with missing values. In the real world, datasets have a high probability of having incomplete data, which means that handling missing values is necessary before selecting relevant features. This renders the results of FS uncertain when the dataset has incomplete data.”..

In addition, we mentioned in the discussion section “This study has several limitations, and the results of the quality of imputation for each method are limited to the used datasets. Hence, the researchers should study their datasets to decide which method to apply. In this sense, the main contribution of our research is not providing a universal solution to handle missing values or select relevant features. Instead, it is showing evidence about the need to consider the impact of missing values in the feature selection process”

(2)The motivation of this research must be clearly and elaborately described. Why this research is important? Currently, a brief discussion is provided which is not enough. Also, Section 4 (discussion) gives some indication, but that is at the end of the paper. So, some of that discussion can be placed in the literature review section.

Response: thank you for your comment. 

We agree. We have changed part of the introduction section to describe better our motivation. We realized that we did not describe well the motivation of our research and this could cause a misunderstanding. We have added more details in the Introduction section and the description of the discussion to align the content of both sections, as follows:

“According to (13), missing values could be present in the target variable in the context of classification. For example, when a classification or estimation model is evaluated, missing values are imputed in the test data's target variable and the model predicts values for the target variable. However, when a dataset has missing values in the features, we must find a way to handle the missing values and perform preprocessing tasks to get a dataset with complete data. Commonly, missing data are solved by removing the instances or features with missing values or replacing the missing values by using basic mechanisms such as mean, mode, etc. Although these strategies are easy to implement, they change the distribution of the dataset and could bias the subsequences analyzes of Machine Learning, for instance, the Feature Selection or Classification processes. On the one hand, the methods to handle missing values could eliminate from the dataset: (i) relevant features or (ii) instances that reveal the importance of the relevant features. On the other hand, the models could be trained by using only a part of the original data.” 

….

“This research aims to describe how Imputation Data can improve the Feature Selection on datasets with missing data and avoid biasing the dataset. For this, we showed the impact of missing values in the FS process by implementing an algorithm of Imputation Data and evaluating it with different datasets to compare the FS process using datasets without handling missing values versus imputed datasets”

(3)In Section 3.1.1.3 and several places “100%” is used. Such as “MICE's overall accuracy was 100% better than the overall accuracy of mode replacement”. It is not clear why this is used. Please clarify how this wording makes sense.

Response: thank you for your comment. we have rewritten the explanation of the results.

For example, the overall accuracy achieved by MICE was better than the overall accuracy achieved by mode replacement in the 100% of the missing rates. 

(4)In Section 3 – the results section, the authors present a number of graphs and results. However, there is little explanation of the results.

Response: We appreciate your comments. We have added references in the discussion section to tables and figures described in the results section. This is because the results section only shows the results of the evaluation, but the analyzes and comparisons with related works are performed in the discussion section. 

(5)In Section 3 – the results section, a formatting issue is present in the form of “Error! Reference source not found”. This needs to be corrected.

Response: Thank you for your comment. We have corrected the reference. 

(6)Overall the presentation and writing of the paper need some editing. Currently, the results and findings are not easy to understand. Particularly, the concept “the impact of imputation in the feature selection process” is not clearly understood in the paper. Also, a clearer discussion is required for the “the use of multiple datasets of different diseases and the difference in findings based on datasets”.

Response: thanks for your comment. 

We kindly ask you to consider that the motivation of this research is not to propose a new method to impute data or select features, it is to evaluate how the imputation data could change the distribution of the dataset and bias the feature selection process. Considering this, we selected 4 well-known datasets available at the UCI Machine Learning repository. The selected datasets have been used in many previous studies of Feature Selection. 

Additionally, we have added more details in the introduction section to make clearer the motivation of our research:

In addition, In the discussion section, we have added some references to tables and figures presented in the results section to relate the results of the evaluation to the final analysis.

---

## [Decision Letter · Decision Letter 1]

2 Jul 2021

Evaluating the impact of multivariate imputation by MICE in feature selection

PONE-D-20-39195R1

Dear Dr. Mera-Gaona,

We’re pleased to inform you that your manuscript has been judged scientifically suitable for publication and will be formally accepted for publication once it meets all outstanding technical requirements.

Kind regards,

Zaher Mundher Yaseen

Academic Editor

PLOS ONE

Additional Editor Comments (optional):

Reviewers' comments:

Reviewer's Responses to Questions

**Comments to the Author**

1. If the authors have adequately addressed your comments raised in a previous round of review and you feel that this manuscript is now acceptable for publication, you may indicate that here to bypass the “Comments to the Author” section, enter your conflict of interest statement in the “Confidential to Editor” section, and submit your "Accept" recommendation.

Reviewer #1: All comments have been addressed

Reviewer #2: All comments have been addressed

2. Is the manuscript technically sound, and do the data support the conclusions?

Reviewer #1: Yes

Reviewer #2: Yes

3. Has the statistical analysis been performed appropriately and rigorously? 

Reviewer #1: Yes

Reviewer #2: Yes

4. Have the authors made all data underlying the findings in their manuscript fully available?

Reviewer #1: Yes

Reviewer #2: Yes

5. Is the manuscript presented in an intelligible fashion and written in standard English?

Reviewer #1: Yes

Reviewer #2: Yes

6. Review Comments to the Author

Reviewer #1: The authors have addressed all my concerns, and I have no more comments. I recommend for acceptance.

Reviewer #2: After the revision, the manuscript has been improved. The authors may wish to elaborate the acronym "MICE" in the abstract and Introduction sections.

7. PLOS authors have the option to publish the peer review history of their article (what does this mean?). If published, this will include your full peer review and any attached files.

Reviewer #1: No

Reviewer #2: No

---

## [Editor Report · Acceptance letter]

16 Jul 2021

PONE-D-20-39195R1 

Evaluating the impact of multivariate imputation by MICE in feature selection 

Dear Dr. Mera-Gaona:

I'm pleased to inform you that your manuscript has been deemed suitable for publication in PLOS ONE. Congratulations! Your manuscript is now with our production department. 

Kind regards, 

on behalf of

Dr. Zaher Mundher Yaseen 

Academic Editor

PLOS ONE